# Iterative HOMER with uncertainties

Anja Butter[1,2], Ayodele Ore[1*], Sofia Palacios Schweitzer[1§], Tilman Plehn[1,3],

and

Benoît Assi[4], Christian Bierlich[5], Phil Ilten[4], Tony Menzo[4], Stephen Mrenna[4,6],
Manuel Szewc[4,7‡], Michael K. Wilkinson[4], Ahmed Youssef[4], and Jure Zupan[4]
(*MLhad collaboration*)

**1** Institut für Theoretische Physik, Universität Heidelberg, Germany
**2** LPNHE, Sorbonne Université, Université Paris Cité, CNRS/IN2P3, Paris, France
**3** Interdisciplinary Center for Scientific Computing (IWR), Universität Heidelberg, Germany
**4** Department of Physics, University of Cincinnati, Cincinnati, Ohio 45221,USA
**5** Department of Physics, Lund University, Box 118, SE-221 00 Lund, Sweden
**6** Computational Science and AI Directorate, Fermilab, Batavia, Illinois, USA
**7** International Center for Advanced Studies (ICAS), ICIFI and ECyT-UNSAM, 25 de Mayo y
Francia, (1650) San Martín, Buenos Aires, Argentina

*a.ore@thphys.uni-heidelberg.de, §palacios@thphys.uni-heidelberg.de, ‡szewcml@ucmail.uc.edu

September 24, 2025

## Abstract

We present `iHOMER`, an iterative version of the `HOMER` method to extract Lund fragmentation functions from experimental data. Through iterations, we address the information gap between latent and observable phase spaces and systematically remove bias. To quantify uncertainties on the inferred weights, we use a combination of Bayesian neural networks and uncertainty-aware regression. We find that the combination of iterations and uncertainty quantification produces well-calibrated weights that accurately reproduce the data distribution. A parametric closure test shows that the iteratively learned fragmentation function is compatible with the true fragmentation function.

# 1 Introduction

Monte Carlo event generators (MCEGs) are essential tools for analysis of collider measurements, providing theory predictions in the form of simulated events [1, 2]. Their accuracy and precision are critical for measurements at experiments such as the Large Hadron Collider (LHC). A key component of MCEGs is the modeling of hadronization — the binding of partons into hadrons, which are then observed in the experiments. Since the process of hadronization is non-perturbative, empirical models are employed to capture the relevant phenomenology [3–5].

Current state of the art generators such as PYTHIA [6], HERWIG [7] and SHERPA [8] employ either one or both of the two main empirical models: the Lund string model [9, 10] and the cluster model [11–13]. The level of precision reached by or obtainable at current and future colliders however, is now such that hadronization has become a relevant systematic uncertainty in many precision measurements, such as the top quark mass measurements, $\alpha_s$ determination from shape observables, or in investigating jet substructure [14–16]. These uncertainties will only become more relevant as modern, unbinned analyses are performed, pushing the limits of MCEG accuracy.

An alternative approach to improving MCEG accuracy is the use of Machine Learning (ML)-based hadronization models [17–23]. Their adoption faces three distinct challenges. First, training of an accurate neural network, or any other hadronization model, is difficult due to the complex relationship between the microscopic dynamics of hadronization and the observable data. Second, any hadronization model, and therefore also the corresponding neural network, should come with reliable uncertainties. Finally, on the more fundamental side, we want to improve our physical description and understanding of hadronization, not just the accuracy of the simulator. The question is how we can leverage ML without discarding physically motivated fragmentation models.

To address these challenges, Refs. [17–23] adopted a modular approach, using ML to replace only a physics-defined and interpretable component of the simulation pipeline. In particular, the HOMER method [22, 23] relies on the Lund string fragmentation picture and learns a fragmentation function $f(z|m_T^2)$ by reweighting a reference distribution $f_{\text{ref}}(z|m_T^2)$, in order to obtain better agreement with data. The strategy has been shown to work for a simplified hadronization scenario involving only pion production, with and without the addition of gluons from a parton shower [22, 23].

In this work, we extend the HOMER method both in accuracy and in precision. Regarding accuracy, we show that the HOMER method can be systematically improved through an iterative procedure that removes any lingering bias due to the factorization assumption at the event level. Regarding precision, we account for the statistical uncertainties of the HOMER method, due to finite training datasets, and for the systematic uncertainties, due to model biases derived from architectural choices and training performance. The end result is an iterative iHOMER variant, based on Bayesian Neural Networks (BNNs) [24–27] and uncertainty-aware regression, where the learned calibrated uncertainties [28,29] can be propagated downstream in the form of weight variations.

The paper is organized as follows. We start by providing a brief review of the Lund string fragmentation model in section 2. We then review in section 3 the HOMER method and introduce the necessary modifications to include uncertainty quantification and the iterative debiasing framework. Section 4 contains details about the simulated dataset, the results are shown in Section 5, and we list our conclusions in Section 6. Further details on the HOMER method are relegated to Appendix A, training specifications can be found in Appendix B, while Appendix C contains results obtained using BNN weights.

## 2 String fragmentation

The Lund string fragmentation model [9,10] is based on the identification of the confined color field between a color-charge and an anti-color-charge with a massless relativistic string. The simplest system one can consider is that of a quark and an anti-quark produced from a color-singlet, e.g. a $Z$-boson produced in an $e^+e^-$ collision, with no gluons present. The string has a constant energy density $\kappa \approx 1$ GeV/fm. When the two string ends move apart, making the string longer, energy will be transferred from the end-points to the string. If the total energy of the initial system is small, this will result in a "yo-yo" motion back and forth. If the energy is large, however, it will at some point become energetically favorable for the string to fragment into smaller pieces — the hadrons. In the simplest case, we restrict ourselves to motion in one spatial dimension and only one quark flavor. It can be shown in this 1+1 dimensional case [9] that the probability of a final state with momenta $(p_1, \cdots, p_n)$ is proportional to the imaginary part of the action of a massless relativistic string with area $A$:

$$\mathcal{P} \propto \left\{ \left[ \prod_{i=1}^n d^2 p_i \delta(p_i^2 - m^2) \right] \delta^{(2)} \left( p_{\text{tot}} - \sum_i p_i \right) \right\} \exp(-bA) \,. \tag{1}$$

where $p_{\text{tot}}$ is the total momentum. Generating this final state from a color-singlet initial state requires several modeling and implementation choices. The key choices relevant to this study are outlined below, followed by a brief overview of the reweighting method.

### 2.1 Model choices

The mechanism for string breaking is the production of $q\bar{q}$-pairs from vacuum fluctuations. Starting from string endpoint quark $q_1$ and producing a pair $q_2\bar{q}_2$, the resulting meson consists of $q_1\bar{q}_2$, while the remaining string now ends with quark $q_2$. The size of the potential barrier to tunnel through is given by the energy of the $q\bar{q}$-pair that is to be created, with the energy available proportional to $\kappa$. The tunneling probability is given by [30],

$$\frac{d\mathcal{P}}{d^2 \Delta \vec{p}_\perp} \propto \exp\left( -\frac{\pi m_{T,q}^2}{\kappa} \right), \tag{2}$$

where $m_{T,q}^2 = m_q^2 + |\Delta \vec{p}_{T,q}|^2$ is the transverse mass of the produced quark; the transverse momentum arises when extending the model from 1+1 to 3+1 dimensions. The produced hadron receives a fraction $z$ of the string's light-cone momentum,

$$z \equiv \frac{(E \pm p_z)_{\text{had}}}{(E \pm p_z)_{\text{string}}}, \tag{3}$$

where $+(-)$ applies to the breaks occurring on the positive (negative) string endpoint. After a hadron takes a fraction $z$ of the string's light-cone momentum, only $(1-z)$ remains for the next iteration. The fraction $z$ is sampled from the symmetric Lund fragmentation function,

$$f(z|m_T^2) \propto \frac{(1-z)^a}{z} \exp\left( -\frac{bm_T^2}{z} \right), \tag{4}$$

where $m_T$ is now the transverse mass of the hadron and $f$ is normalized to 1 in $z \in [0,1]$. The fragmentation function in Eq.(4) is the simplest form derived from a set of minimal assumptions about the fragmentation process [30], such as left-right symmetry and causality.

In this work, we will restrict ourselves to the above problem of hadronizing $q\bar{q}$ strings without gluons, and further restrict the final states to consist exclusively of pions. We do this both to

compare with the validated results of [22] and to keep the complications of the model to a minimum when introducing the iterative procedure and the uncertainty quantification. The Lund model as such is, however, able to handle more complex topologies involving gluons [30–32], production of baryons [33–35], and string interactions in multi-string configurations [36–38], as well as hadronizing junction topologies emerging from string interactions or hadronic beam remnants [39–41].

## 2.2 Implementation choices

The Lund string model is implemented in the PYTHIA Monte Carlo event generator [6], and has remained a core component since its origin as JETSET over 40 years ago. Its implementation introduces additional choices, of which the most relevant for this study are outlined below.

The produced hadrons are spacelike separated, and thus causally disconnected, meaning there is no preferred time ordering in the fragmentation process. In PYTHIA, this is implemented as an "outside-in" cascade, starting from the $q\bar{q}$ string ends and then moving inward. This introduces two implementation choices. First, the algorithm randomly selects a string-end to hadronize at each step. This choice is encoded as a binary feature fromPos $\in \{\pm 1\}$ which selects the sign in Eq.(3). Second, the "outside-in" approach leaves the algorithm with a small-mass remnant in the middle, which must be handled. This is done by the finalTwo algorithm, which attempts to generate two on-shell hadrons from the remnant, rejecting the constructed hadronization chain if this is not possible (due to the inability to conserve energy and momentum, while still respecting the Lund string fragmentation function probability distribution).

The model itself does not prescribe whether transverse or longitudinal degrees of freedom should be generated first. The default choice in PYTHIA is to generate flavor and transverse momentum first, allowing for the hadron transverse mass $m_T$ to enter the selection of $z$ from Eq.(4). The algorithm for the emission of a single hadron (a "string break") can thus be summarized as the following.

1. Randomly select fromPos, which determines the string end to be considered.
2. Select a quark flavor. For the purposes of this paper, only $u$ and $d$ quarks are relevant, each chosen with equal probability.
3. Select a hadron from the combination of the original and new flavor. This determines the hadron mass $m_{\text{had}}$.
4. Sample the transverse momentum $\Delta\vec{p}_T$ of the quarks from a Gaussian with a tunable width $\sigma_T \approx 300$ MeV, exploiting the factorization property of Eq.(2). The meson produced in the string break has a total transverse momentum

$$\vec{p}_{T,\text{had}} = \Delta\vec{p}_T + \vec{p}_{T,\text{string}} \, ,$$

while the transverse momentum of the string fragment is updated accordingly,

$$\vec{p}_{T,\text{string}} \rightarrow \vec{p}_{T,\text{string}} - \Delta\vec{p}_T \, .$$

5. Sample the light-cone momentum fraction $z$. The parameter $b$ in Eq.(4) is universal, while the parameter $a$ can be modified depending on quark flavor (not relevant for this paper).

## 2.3 Reweighting fragmentations

In Section 3, we will use the HOMER method to find corrected probabilities for string fragmentation through reweighting. To reweight fragmentations, one needs to specify a probabilistic

model of a string break, which is represented as a set of features (we use a shortened version of the notation from Refs. [22, 23], see Table 1 in App. A)

$$s \equiv \left\{z, \Delta\vec{p}_T, m_{\text{had}}, \texttt{fromPos}, \vec{p}_{T,\text{string}}\right\} . \tag{5}$$

We are primarily interested in the probability conditioned on the string state prior to the break

$$p(s) \equiv p(z, \Delta\vec{p}_T, m_{\text{had}}, \texttt{fromPos}|\vec{p}_{T,\text{string}})$$
$$= f\left(z|m_T^2\right)\mathcal{N}(\Delta\vec{p}_T|0, \sigma_T)p(m_{\text{had}})\text{Bern}(\texttt{fromPos}|0.5) \tag{6}$$

where $\mathcal{N}(\cdot|\mu, \sigma)$ is the multivariate Gaussian distribution with mean $\mu$ and diagonal covariance $\sigma^2\mathbb{1}$, and $\text{Bern}(\cdot|p)$ is the Bernoulli distribution with probability of success $p$. Note that here we abuse notation to define $p(s)$ as the conditional probability of the sampled features given the string state. It is not to be interpreted as the joint probability of all the components of $s$.

A generated sequence of string breaks is called a fragmentation chain, $S \equiv (s_1, \cdots, s_N)$, and follows a factorized probability distribution

$$p(S) = \prod_{s \in S} p(s) . \tag{7}$$

The observable event $x$, specified by the four momenta and flavors of the produced hadrons, is a deterministic projection of the chain $S$

$$x = \mathcal{O}(S). \tag{8}$$

The operator $\mathcal{O}$ projects out the sampled values of $z$, $\texttt{fromPos}$ and $\vec{p}_{T,\text{string}}$ and gives the four momenta of hadrons, out of which one can form observables measurable by experiment. Note that a given chain $S$ leads to a unique observed event $x$, but the opposite is not true. In general, one cannot uniquely reconstruct $\mathcal{O}^{-1}(x)$ since $x$ does not, for example, contain information on the order of string breaks; several different orderings with appropriately modified $z$, $\texttt{fromPos}$ and $\vec{p}_{T,\text{string}}$ lead to the same $x$. Instead, a measured event $x$ is linked to a probability distribution over compatible chains $S$, which we write as the Bayesian posterior

$$p(S|x) = \frac{p(x, S)}{p(x)} = \frac{p(x|S)p(S)}{p(x)} = \frac{\delta(x - \mathcal{O}(S))p(S)}{\int dS\delta(x - \mathcal{O}(S))p(S)} . \tag{9}$$

Simulations provide implicit access to this posterior through generated pairs $(x, S)$.

The length of a chain $S$ is set when the string invariant mass falls below an energy threshold, at which point PYTHIA applies its $\texttt{finalTwo}$ algorithm which attempts to generate two hadrons from the remaining string. In actuality, and due to the iterative nature of the string breaking procedure and the discrete number of possible hadron masses, this may not be kinematically possible and in such a case the chain is rejected and the hadronization starts anew. Thus, a full simulation history $H \equiv \left(S_1^{\text{rej}}, \cdots, S_N^{\text{rej}}, S^{\text{acc}}\right)$ contains also fragmentation chains $S_i^{\text{rej}}$ rejected by the $\texttt{finalTwo}$ algorithm, with only the final fragmentation chain $S^{\text{acc}}$ accepted.

Because an event can only be generated from an accepted chain, the posterior in Eq.(9) is defined only over the subset of accepted chains $S^{\text{acc}}$. To correctly compute the probability of an event using just the probabilities for the accepted chains, we need to take into account the acceptance rate

$$\alpha \equiv \int dS^{\text{acc}}p(S^{\text{acc}}) = \int dS\mathcal{A}(S)p(S) = \langle\mathcal{A}(S)\rangle_{p(S)} , \tag{10}$$

$$\text{with} \quad \mathcal{A}(S) = \begin{cases} 1 & S \text{ accepted} \\ 0 & S \text{ rejected} . \end{cases}$$

We can then write

$$p(x) = \frac{\int dS^{\mathrm{acc}} \, p(x|S^{\mathrm{acc}})p(S^{\mathrm{acc}})}{\int dS^{\mathrm{acc}} \, p(S^{\mathrm{acc}})} = \frac{\int dS^{\mathrm{acc}} \, \delta(x - \mathcal{O}(S^{\mathrm{acc}}))p(S^{\mathrm{acc}})}{\alpha}, \tag{11}$$

where the integration is restricted to accepted chains $S^{\mathrm{acc}}$, see App. A for the derivation. The posterior $p(S^{\mathrm{acc}}|x)$ does not explicitly depend on $\alpha$ because of a cancellation in the ratio between the joint $p(x, S^{\mathrm{acc}})$ and the marginal $p(x)$. The acceptance rate can be estimated using a set of $M$ simulated histories $H$ (and thus $M$ accepted chains), by counting how many chains were generated in total,

$$\alpha \approx \frac{M}{\sum_{m=1}^{M} (N_m + 1)}, \tag{12}$$

where $N_m$ is the number of rejected chains in the $m^{\mathrm{th}}$ simulated history and there is always one accepted chain per history. Beyond being used to compute the acceptance rate, histories contain the complete information of the generated samples and are used to compute the best possible weights that can be achieved in a closure test.

## 3   Reweighting with HOMER

HOMER is a method to extract an optimal fragmentation function by reweighting a PYTHIA reference simulation to match experimental data [22, 23]. Its goal is to approximate the exact single emission weights $w(s)$,

$$w(s) \equiv \frac{p_{\mathrm{data}}(z, \Delta\vec{p}_T, m_{\mathrm{had}}, \texttt{fromPos}|\vec{p}_{T,\mathrm{string}})}{p_{\mathrm{ref}}(z, \Delta\vec{p}_T, m_{\mathrm{had}}, \texttt{fromPos}|\vec{p}_{T,\mathrm{string}})}, \tag{13}$$

where $p_{\mathrm{data}}$ refers to the data probability distribution and $p_{\mathrm{ref}}$ to the reference distribution, given by the choice of simulator parameters. Corresponding weights for chains and histories follow by taking products

$$w(S) \equiv \frac{p_{\mathrm{data}}(S)}{p_{\mathrm{ref}}(S)} = \prod_{s \in S} w(s) \qquad \text{and} \qquad w(H) \equiv \frac{p_{\mathrm{data}}(H)}{p_{\mathrm{ref}}(H)} = \prod_{S \in H} w(S). \tag{14}$$

Throughout this reweighting procedure, HOMER still manifestly assumes and respects the Lund string fragmentation picture, including momentum conservation, and thus delegates most of the physics constraints to PYTHIA without explicitly hard-coding them.

The conceptual and practical challenge is that experimental observations do not have access to individual string breaks, but are rather functions of only the final hadron momenta. In HOMER, this problem is solved by splitting the training procedure into two steps:

**Step 1 (Reweighting):**  A classifier is trained to distinguish experimental measurements from predictions of a reference PYTHIA simulation. This yields an approximation to the likelihood ratio $p_{\mathrm{data}}(x)/p_{\mathrm{ref}}(x)$.

**Step 2 (Factorization):**  Using the reference simulation only, a regressor operating on string-break features is trained to statistically match the Step 1 density ratio at the observable level. This yields inferred string-break weights.

`HOMER` has been shown to work for $q\bar{q}$ strings with and without parton showers [22, 23], but some challenges remain. Among them is uncertainty quantification. Generally speaking, we expect neural network training to be subject to both statistical and systematic uncertainties [28]. Statistical uncertainties capture the effect of having training datasets of finite size, while systematic uncertainties remain for arbitrary training statistics and capture, for instance, noisy data, insufficient network expressivity, or a limiting training strategy. For `HOMER`, we can make the following assumptions:

1. The classifier is sufficiently expressive to fit the true likelihood ratio $p_{\text{data}}(x)/p_{\text{ref}}(x)$
   $\implies$ We consider dominantly statistical uncertainties in the reweighting Step 1.
2. A sufficiently large reference dataset is available
   $\implies$ We consider only systematic uncertainties in the factorization Step 2.

Regarding the first assumption, we check that no systematic biases arise due to architectural choices or imperfect network training. Additionally, because we are performing a closure test using simulated datasets, we know that the simulator perfectly captures all details of the data, except for the targeted hadronization model, and we can safely neglect systematic uncertainties due to simulator mismodeling of non-hadronization effects. We assume that for the correct hadronization model $x = \mathcal{O}(S^{\text{acc}})$ induces the correct distribution in observable space.

As for the second assumption, since each event contains approximately $\mathcal{O}(15)$ fragmentations, the effective statistics of the reference simulation used to learn the fragmentation function in Step 2 is the corresponding order of magnitude larger than the number of generated events. As a result, we find that statistical uncertainties can be neglected.

Given these assumptions, our approach to quantifying fragmentation uncertainties is to train Step 1 as a Bayesian Neural Network (BNN) [24, 25, 42, 43], to capture the statistical uncertainties, and Step 2 with a heteroscedastic loss to capture the systematics. In this combination, the uncertainty learned for Step 1 propagates to the systematics in Step 2 as label noise, and should be captured by the learned uncertainty. Here the strict separation of the two sources of uncertainties should be taken with a grain of salt, because the BNN classifier will still attempt to accommodate systematics, if they occur, and the heteroscedastic loss can improve the training also for statistical limitations [28].

## 3.1 Step 1: Event reweighting

Step 1 of `HOMER` estimates the data-reference likelihood ratio over the observable phase space,

$$w(x) \equiv \frac{p_{\text{data}}(x)}{p_{\text{ref}}(x)} \,. \tag{15}$$

We estimate it via a trained classifier $C_\theta(x)$, to distinguish events sampled from $p_{\text{data}}(x)$ and $p_{\text{ref}}(x)$. To quantify statistical uncertainties, we employ a BNN [44]. This involves a variational approximation to the posterior distribution of the network parameters $\theta$ conditioned on the training dataset [44],

$$q(\theta) \approx p(\theta|\mathcal{D}_{\text{train}}) \,. \tag{16}$$

Here, $\mathcal{D}_{\text{train}}$ consists of samples from $p_{\text{data}}(x)$ and $p_{\text{ref}}(x)$ with corresponding binary labels. The BNN loss maximizes the corresponding likelihood and, from Bayes' theorem, also includes a KL-divergence between $q(\theta)$ as a regularization and a finite-width prior $p(\theta)$

$$\mathcal{L}_{\text{class}} = D_{\text{KL}}[q(\theta), p(\theta)] - \left\langle \log p(\mathcal{D}_{\text{train}}|\theta) \right\rangle_{q(\theta)}$$
$$= D_{\text{KL}}[q(\theta), p(\theta)] - \left\langle N_{\text{data}} \left\langle \log C_\theta(x) \right\rangle_{p_{\text{data}}(x)} + N_{\text{ref}} \left\langle \log[1 - C_\theta(x)] \right\rangle_{p_{\text{ref}}(x)} \right\rangle_{q(\theta)} \,. \tag{17}$$

For the classification we choose a binary cross entropy likelihood. To be able to compute the KL divergence, we choose

$$q(\theta) \equiv q_{\mu,\sigma}(\theta) \tag{18}$$

and $p(\theta)$ to be independent Gaussian distributions for each network parameter. For $p(\theta)$ the means are fixed to zero, while the standard deviations are set to a constant value which is a hyperparameter. The BNN network parameters consist of the means $\mu$ and standard deviations $\sigma$ for each weight. This effectively doubles the number of network parameters. We emphasize that for deep networks, Gaussian weights do not imply Gaussian network output [26, 45, 46].

The independent Gaussians allow for efficient sampling during training and for uncertainty estimation. For a given point $x$ in phase space, we can sample $\theta \sim q(\theta)$ and compute the classifier output $C_\theta(x)$. In the limit of perfect training, and assuming $N_{\text{data}} = N_{\text{ref}}$, each sampled classifier $C_\theta$ is related to the likelihood ratio by

$$\frac{C_\theta(x)}{1 - C_\theta(x)} \equiv w_\theta(x) \approx w(x), \tag{19}$$

Throughout this paper, weights with Greek subscripts represent NN parameterizations, while the absence of a subscript indicates an exact density ratio.

Without assuming a Gaussian shape, we can estimate the expected value of any quantity and its uncertainty by drawing samples from the BNN and taking the sample mean and standard deviations. Of particular interest is the variance of the log-weight

$$\sigma_q^2(x) \equiv \text{Var}_{q(\theta)}\left[\log w_\theta(x)\right]. \tag{20}$$

We will use it to validate the results of Step 2.

## 3.2 Step 2: Weight factorization

The goal of Step 2 in HOMER is to approximate the string-break level weights $w(s)$, defined in Eq.(13). As in Refs. [22, 23], the only difference between data and simulation lies in the choice of fragmentation function $f(z|m_T^2)$. The string-break level weights are then given by

$$w(s) \equiv \frac{p_{\text{data}}(z, \Delta\vec{p}_T, m_{\text{had}}, \texttt{fromPos}|\vec{p}_{T,\text{string}})}{p_{\text{ref}}(z, \Delta\vec{p}_T, m_{\text{had}}, \texttt{fromPos}|\vec{p}_{T,\text{string}})} = \frac{f_{\text{data}}(z|m_T^2)}{f_{\text{ref}}(z|m_T^2)}, \tag{21}$$

and depend only on $z$ and $m_T$. We emphasize, however, that this simplification is not assumed during training and the full $s$ can be kept as the input.

While the Step 2 network is trained on a simulated reference dataset, the problem is constrained by data through the trained Step 1 classifier $w_\theta(x)$. To allow for this link, we write the event-level weights induced by $w(s)$ as

$$w(x) = \frac{\alpha_{\text{ref}}}{\alpha_{\text{data}}} \left\langle \prod_{s \in S^{\text{acc}}} w(s) \right\rangle_{p_{\text{ref}}(S^{\text{acc}}|x)}, \tag{22}$$

where $p_{\text{ref}}(S^{\text{acc}}|x)$ is defined as in Eq.(9), with the reference label added to specify the model, and evaluated only over the subset of accepted chains. We introduce acceptance efficiencies $\alpha_{\text{ref}}$ and $\alpha_{\text{data}}$ for the reference and data, respectively.

While Eq.(22) constitutes an exact relation between string break-level and event-level weights, the averaging it requires is not practical to implement. This is because the probability that multiple accepted chains in a simulation produce the exact same event is negligible,

in other words, the posterior $p_{\text{ref}}(S^{\text{acc}}|x)$ is always very "sharp". This implies that averaging requires arbitrarily large simulated samples. The simplest approximation is to consider only the accepted chain used to generate each event such that

$$w(x) \approx \frac{\alpha_{\text{ref}}}{\alpha_{\text{data}}} \prod_{s \in S^{\text{acc}}} w(s) \Bigg|_{x=\mathcal{O}(S^{\text{acc}})} . \tag{23}$$

This approximation, which would be exact when the observables were invertible and $S^{\text{acc}}$ could be uniquely assigned, avoids explicit averaging when relating event- and chain-level weights. However, when computing binned observable distributions, an implicit averaging is performed by integrating over all events belonging to the same bin.

The approximate Eq.(23) is used in Step 2 of HOMER, where we train a fragmentation-level network $w_\phi(s)$, with parameters $\phi$, so that it satisfies

$$w_\theta(x) \approx \frac{\alpha_{\text{ref}}}{\alpha(\phi)} \prod_{s \in S^{\text{acc}}} w_\phi(s) \equiv \frac{\alpha_{\text{ref}}}{\alpha(\phi)} w_\phi(S^{\text{acc}}) . \tag{24}$$

In the above, $\alpha(\phi)$ is the acceptance rate induced by weighting the reference with $w_\phi(s)$,

$$\alpha(\phi) = \int dS \mathcal{A}(S) w_\phi(S) p_{\text{ref}}(S) = \left\langle \mathcal{A}(S) w_\phi(S) \right\rangle_{p_{\text{ref}}(S)} . \tag{25}$$

It is needed to ensure correct normalization of the weights, and can be approximated using histories sampled from the reference simulation as

$$\alpha(\phi) \approx \frac{\sum_{S^{\text{acc}}} w_\phi(S^{\text{acc}})}{\sum_{S^{\text{rej}}} w_\phi(S^{\text{rej}}) + \sum_{S^{\text{acc}}} w_\phi(S^{\text{acc}})} , \tag{26}$$

where the denominator gives the effective number of both accepted and rejected chains. In effect, the matching procedure of Eq.(24) factorizes the classifier weight $w_\theta(x)$ as a product of string break-level weights $w_\phi(s)$. If both Step 1 and Step 2 of HOMER are optimally trained, we expect the learned weights to approximate the exact weights at the fragmentation level, $w_\phi(s) \approx w(s)$, subject to the error associated with the approximation in Eq.(23).

The matching in Eq.(24) can be enforced in several ways. For instance, Ref. [22] trains $w_\phi(s)$ with another BCE loss. Since we aim to simultaneously learn systematic uncertainties, we use a heteroscedastic regression loss. This amounts to fitting the learned distribution of Step 1 BNN log-weights with a Gaussian likelihood that has a learnable mean and standard deviation,

$$\mathcal{L}_{\text{factor}}(\phi) = -\left\langle \log \mathcal{N}\left( \log w_\theta(x = \mathcal{O}(S^{\text{acc}})) \middle| \log \frac{\alpha_{\text{ref}}}{\alpha(\phi)} \prod_{s \in S^{\text{acc}}} w_\phi(s), \sigma_\phi(S^{\text{acc}}) \right) \right\rangle_{p_{\text{ref}}(S^{\text{acc}}), q(\theta)} . \tag{27}$$

We consider the log-weights because they can be negative, unlike the positive-definite weights, and tend to be more naturally modeled by a simple Gaussian. The expectation is taken over paired events and accepted chains, where $x = \mathcal{O}(S^{\text{acc}})$, and over Step 1 classifier parameters $\theta$ sampled from the BNN. The rejected chains of the simulated histories are taken into account when computing the acceptance rates via Eq.(26). The expected value in the loss is constructed in terms of individual fragmentation-level log-weights $\log w_\phi(s)$ through Eq.(24) in its logarithmic form, and the uncertainty $\sigma_\phi(S^{\text{acc}})$ is likewise constructed from individual fragmentation-level uncertainties by adding in quadrature

$$\sigma_\phi^2(S^{\text{acc}}) \equiv \sum_{s \in S^{\text{acc}}} \sigma_\phi^2(s) . \tag{28}$$

This ignores the uncertainty contribution from the acceptance rate $\alpha(\phi)$, which we have verified to be negligible by injecting noise to the fragmentation weights and observing insignificant changes in the rate. For the same reason, we also do not consider Jensen's inequality when constructing the expected data acceptance rate $\alpha(\phi)$ in terms of the exponential of the expected log-weights.

Although we denote $w_\phi(s)$ and $\sigma_\phi(s)$ with the same parameter label $\phi$, these can be implemented either as two outputs of a single network or as two independent networks. In the numerical results below, we keep the networks separate. Additionally, we modify the inputs of the networks from the full set of string break features to

$$s \rightarrow \left\{ z, \vec{p}_{T,\text{string}} + \Delta\vec{p}_T, m_{\text{had}}, \texttt{fromPos} \right\}. \tag{29}$$

Without this feature reduction, the networks can learn to exploit the arbitrary distinction between first and subsequent fragmentations in a chain, where the former always have $\vec{p}_{T,\text{string}} = 0$, and the ultimate performance is compromised.

By construction, successful training with the loss (27) ensures that log-weight predictions are calibrated against the classifier, meaning that the pull statistic

$$t\left[w_\theta(x) \,|\, \alpha_{\text{ref}}/\alpha(\phi)w_\phi(S^{\text{acc}}), \sigma_\phi(S^{\text{acc}})\right] \equiv \frac{\log w_\theta(x) - \log \dfrac{\alpha_{\text{ref}}}{\alpha(\phi)} w_\phi(S^{\text{acc}})}{\sigma_\phi(S^{\text{acc}})}, \tag{30}$$

is normally distributed when sampling $p_{\text{ref}}(S^{\text{acc}})$ (with $x = \mathcal{O}(S^{\text{acc}})$) and $q(\theta)$. Even in a realistic application to data, this pull can always be checked in order to validate that $\sigma_\phi$ has been learned correctly. Ultimately, we hope this translates into calibration at the fragmentation level so that the pull $t[w(s)|w_\phi(s), \sigma_\phi(s)]$, defined analogously to Eq.(30), is normally distributed when sampling $p_{\text{ref}}(s)$. However, as we detail below, this may not be exactly the case due to the observable information failing to completely constrain the fragmentation-level weights.

To summarize, the output of HOMER is now a string-break-level log-weight and its uncertainty, $\log w_\phi(s) \pm \sigma_\phi(s)$. This induces corresponding expected log-weights and uncertainties at the chain, and history levels,

$$\log w_\phi(S) = \left(\sum_{s \in S} \log w_\phi(s)\right) \pm \sqrt{\sum_{s \in S} \sigma_\phi^2(s)}$$

$$\log w_\phi(H) = \left(\sum_{S \in H} \log w_\phi(S)\right) \pm \sqrt{\sum_{S \in H} \sigma_\phi^2(S)}. \tag{31}$$

We do not consider an additional uncertainty associated with $\alpha(\phi)$ when computing the event level log-weights $\log \frac{\alpha_{\text{ref}}}{\alpha(\phi)} w_\phi(S^{\text{acc}})$. The learned expected log-weights and uncertainties can then be translated to expected weights with uncertainties through the first two moments of the log-normal distribution,

$$\langle w \rangle_{\log w \sim \mathcal{N}(\log w_\phi, \sigma_\phi)} = \exp(\log w_\phi + \sigma_\phi^2/2)$$

$$\langle w^2 \rangle_{\log w \sim \mathcal{N}(\log w_\phi, \sigma_\phi)} = \exp(2\log w_\phi + 2\sigma_\phi^2). \tag{32}$$

### 3.3 Iterative HOMER

When introducing HOMER, Ref. [22] found promising results for the fragmentation function, but the event-level observables showed a bias towards the reference simulation. This bias reflects a conceptual limitation due to the main assumption of Eq.(11), that the event weight

factorizes in the same way as the chain weight (modulo the acceptance ratio). This is generally not the case. In part to address this issue, Ref. [23] introduced an alternative strategy that approximates the expectation value in Eq.(22) using a smearing kernel of tunable width and averaging over all histories in a given batch. While effective, the kernel increases the numerical requirements of the algorithm and adds a hyperparameter to be tuned. Additionally, the introduction of a smearing kernel increases the variance of the weights and reduces the effective sample size of the reweighted samples.

An alternative strategy to improve HOMER for non-invertible observables is inspired by iterative unfolding [47–49]. There is also a correspondence with expectation-maximization [50], where the unseen chain-level weights are the latent variables whose expectation value is approximated iteratively. Applied to HOMER, we propagate the inferred fragmentation weights forward to another round of inference.

We introduce two strategies to iterate HOMER. They differ only in Step 2, where one strategy refines the result of the previous iteration, while the other strategy restarts from the initial reference distribution. In most cases, we observe the same behavior and final performance from both iteration styles but believe it might be useful to still consider both strategies. For the results presented in Section 5 we use the first, "refinement" strategy.

**Refinement-style iteration**

At the end of iteration $i$, which produced Step 2 weights $w_{\phi_i}(s)$, we freeze the parameters $\phi_i$ and update the reference distribution to

$$p_{\text{ref}}^{(i+1)}(s) = w_{\phi_i}(s) p_{\text{ref}}^{(i)}(s) \,. \tag{33}$$

The next iteration of Step 1 and Step 2 training proceeds through the same loss functions introduced above, but with the replacement $p_{\text{ref}} \to p_{\text{ref}}^{(i)}$. Applying Eq.(33) recursively, we can write the reference distribution at iteration $i \geq 1$ as

$$p_{\text{ref}}^{(i)}(s) = \left( \prod_{j=1}^{i-1} w_{\phi_j}(s) \right) p_{\text{ref}}(s) \,, \tag{34}$$

where $p_{\text{ref}}^{(1)} = p_{\text{ref}}$ is the original reference distribution. The induced distributions at the chain and event levels are

$$p_{\text{ref}}^{(i)}(S) = \left( \prod_{j=1}^{i-1} w_{\phi_j}(S) \right) p_{\text{ref}}(S)$$

$$p_{\text{ref}}^{(i)}(x) = \left( \prod_{j=1}^{i-1} \frac{\alpha(\phi_{j-1})}{\alpha(\phi_j)} w_{\phi_j}(S^{\text{acc}}) \right) p_{\text{ref}}(x) \,, \tag{35}$$

where we make use of the pairing between $x$ and $S^{\text{acc}}$ in simulation in the second line and understand $\alpha(\phi_0) = \alpha_{\text{ref}}$. At the end of $N$ iterations, the cumulative string-break weights are

$$w_\phi(s) = \prod_{i=1}^{N} w_{\phi_i}(s) \,. \tag{36}$$

The chain and history weights $w_\phi(S)$ and $w_\phi(H)$ are built from $w_\phi(s)$ by taking products according to their usual definitions (31), while the event weights need only the first and last acceptance rates $\alpha$,

$$\left( \prod_{i=1}^{N} \frac{\alpha(\phi_{i-1})}{\alpha(\phi_i)} w_{\phi_i}(S^{\text{acc}}) \right) = \frac{\alpha_{\text{ref}}}{\alpha(\phi_N)} \prod_{i=1}^{N} w_{\phi_i}(S^{\text{acc}}) = \frac{\alpha_{\text{ref}}}{\alpha(\phi_N)} w_\phi(S^{\text{acc}}) \,. \tag{37}$$

**Restart-style iteration**

Instead of updating the Step 2 reference distribution as described above, we can alternatively train Step 2 from scratch using the updated Step 1 results. While Step 1 remains unchanged with respect to the previous iteration style, using Eq.(33) to define $p_{\text{ref}}^{(i)}(x)$ and train the classifier at iteration $i$, the reference distribution for Step 2 is always the original

$$p_{\text{ref}}^{(i)}(s) = p_{\text{ref}}(s) \, . \tag{38}$$

In Step 2 of iteration $i$, the regressor is trained to estimate

$$\frac{\alpha_{\text{ref}}}{\alpha(\phi_i)} w_{\phi_i}(S^{\text{acc}}) \approx \frac{\alpha_{\text{ref}}}{\alpha(\phi_{i-1})} w_{\phi_{i-1}}(S^{\text{acc}}) w_{\theta_i}(x) \, , \tag{39}$$

that is, an updated version of the previous Step 2 prediction including a Step 1 classifier that corrects for non-closure in event-level observables. The corresponding loss is

$$\mathcal{L}_{\text{factor}}^{(i)}(\phi) = \tag{40}$$
$$-\left\langle \log \mathcal{N}\left( \log \frac{\alpha_{\text{ref}}}{\alpha(\phi_{i-1})} w_{\phi_{i-1}}(S^{\text{acc}}) + \log w_\theta(x) \Big| \log \frac{\alpha_{\text{ref}}}{\alpha(\phi_i)} w_{\phi_i}(S^{\text{acc}}), \sigma_{\phi_i}(S^{\text{acc}}) \right) \right\rangle_{p_{\text{ref}}(S^{\text{acc}}), q_i(\theta)} \, ,$$

which differs from Eq.(27) only by the addition of $\log \frac{\alpha_{\text{ref}}}{\alpha(\phi_{i-1})} w_{\phi_{i-1}}$. Since we learn the fragmentation function weight from scratch at each iteration, the final Step 2 result will simply be the output of the regression network of the last iteration. This is the reason we call this alternative training strategy the "restart" iteration style. For efficiency, we find it beneficial to initialize the Step 2 regressor with the parameters of the previous iteration.

**Stopping condition**

Iterating will gradually improve the agreement between the iHOMER predictions and data at an observable level. However, the variance of the weights increases with each iteration due to the accumulation of training uncertainties. To avoid introducing unnecessary noise into training, and to reduce computational costs, we only apply a minimal number of iterations.

A stopping condition requires a performance metric. Here, we discuss a number of possibilities before introducing our metric of choice used in the remainder of this work, the area-under-the-curve (AUC) from the Step 1 classifier at iteration $i$. Since HOMER distills the Step 1 classifier into a learned fragmentation function, this metric should be defined in terms of the agreement between the observed distribution and the Step 2-derived distribution. However, computing the goodness-of-fit of a model defined over a multi-dimensional unbinned distribution is far from trivial [51].

One possible approach is to bin each observable and compute a global $\chi^2/N_{\text{bins}}$. However, this approach may fail to capture subtle differences between non-perturbative models, if perturbative physics dominates. Due to this, one may want to concentrate on the agreement in the features most sensitive to fragmentation, such as the full and charged multiplicity distributions, or their moments.

Alternatively, we could use the Step 1 classifier as a summary statistic, and study the agreement between the weight distributions. Since in this work the classifier distinguishes between two simulated datasets that differ only in their non-perturbative parameters, it should be focused on non-perturbative physics. In an application to real data, the perturbative behavior of the classifier score should be studied carefully. If we compare the weight distributions through unbinned tools such as e.g., Kolgomorov-Smirnov test or the Earth Mover's Distance [52],

these metrics will be extremely sensitive to small imperfections in modeling, and may make a threshold definition difficult. Binned probability distribution comparisons, such as the KL divergence or the Hellinger distance [53], are more robust against modeling imperfections but at the expense of introducing a binning choice.

A related possibility is to compare the weight distributions by assessing the performance of the Step 1 classifier with a "global" metric, where the performance of the classifier trained at iteration $i+1$ reflects how the fragmentation model learned at iteration $i$ compares with data at the observable level. The simplest metric is the AUC. Although it may not capture local failure modes [54], it provides a cheap, fast metric with a well-defined target value of 0.5. Visual inspection of the weight distributions validate this simple choice, but future implementations may require less global stopping metrics.

We expect the Step 1 classifier performance to degrade as we iterate, both because the `iHOMER` predictions become more similar to data, and because the effective sample size of the reference distribution,

$$N_{\text{eff}} = N_{\text{ref}} \frac{\left\langle w_\phi(S^{\text{acc}}) \right\rangle^2_{p_{\text{ref}}(S^{\text{acc}})}}{\left\langle w_\phi(S^{\text{acc}})^2 \right\rangle_{p_{\text{ref}}(S^{\text{acc}})}} , \qquad (41)$$

decreases. Even assuming that the classifier is always trained optimally, its AUC should monotonically decrease to 0.5 once the statistical fluctuations in the training samples are comparable to the differences between the reference and data distributions. We stop iterating once the classifier AUC is compatible with 0.5, where we define "compatible" statistically by sampling the BNN posterior. Specifically, we estimate the mean AUC of the classifier, and its associated uncertainty

$$\mu_{\text{AUC}} \equiv \left\langle \text{AUC}[w_\theta] \right\rangle_{q(\theta)} \qquad \text{and} \qquad \sigma^2_{\text{AUC}} \equiv \text{Var}_{q(\theta)}\left[ \text{AUC}[w_\theta] \right], \qquad (42)$$

using $M$ samples, and take the stopping condition to be

$$\mu_{\text{AUC}} - \frac{1}{\sqrt{M}} \sigma_{\text{AUC}} < 0.5 . \qquad (43)$$

It is satisfied in two distinct cases: either the reweighted simulation and the observed data are indistinguishable, or the Step 1 classifier fails to learn the likelihood ratio. In both cases we want to stop iterating. In the latter case, however, we should re-tune the hyperparameters of the given Step 1 classifier and proceed iterating. Therefore, we also require Step 1 weights to match the precision of the previous iteration.

**Uncertainty-aware `iHOMER`**

In Sections 3.1 and 3.2 we introduced uncertainties in Step 1 and 2 of `HOMER`, and these uncertainties can be computed at each iteration of `iHOMER` regardless of the iteration style. At each iteration $i$ we train a BNN $q_i(\theta)$ during Step 1 and estimate a string break uncertainty $\sigma_{\phi_i}(s)$ in Step 2.

When iterating, the reference distribution is re-defined via Eq.(33) using the predicted means for the weights, without explicitly accounting for the uncertainties. This is a valid working assumption, since $p_{\text{ref}}$ is a fixed reference to reweight from, however arbitrary it may be, and it greatly simplifies things. Since the reference is fixed at iteration $i$, we only care about the uncertainties of that iteration associated with reweighting the reference at iteration $i$ in order to match data. We do not need to combine uncertainties across iterations. This statement, which seems natural for the Step 2 weights in the "restart" iteration style,

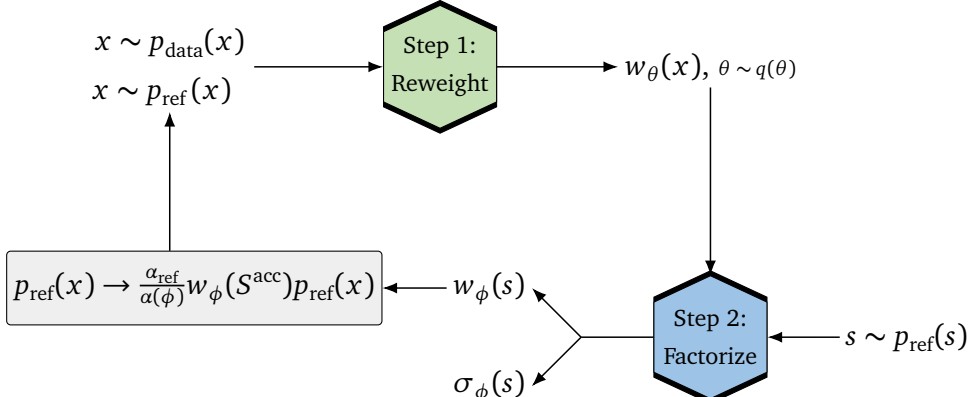

Figure 1: Schematic visualization of `iHOMER`, including uncertainties and iterations. Step 1 and Step 2 networks are defined by their weights $\theta$ and $\phi$ respectively.

also applies to the Step 2 weights in the "refinement" iteration style as well as to the Step 1 weights. We still compute the uncertainties at each iteration, since they are needed for the stopping criterion and because they are useful as means of validation.

The `iHOMER` setup is summarized in Figure 1, showing how uncertainties are introduced for both Steps 1 and 2 through the sampling of network parameters and the additional uncertainty $\sigma_\phi(s)$, respectively. At the end of each iteration, we use the Step 2 results to update $p_{\text{ref}}$ accordingly until the stopping criterion is met.

## 4   Simulated datasets

`HOMER` [20] and its extension to strings with gluons [23] was validated through a closure test where two simulated datasets, with different fragmentation parameters $a$, were used as "Simulation" (a dataset sampled from $p_{\text{ref}}$ and which contains the full simulation history) and "Data" (a dataset sampled from $p_{\text{data}}$ and from which `HOMER` can only access observable events). To validate our uncertainty quantification and the iterative debiasing, we consider slightly modified datasets. The reference dataset, denoted "Simulation", is generated with PYTHIA 8.312 exactly as in Ref. [22] and contains $2 \times 10^6$ events generated from the Lund string fragmentation model with

$$a = 0.68 \qquad b = 0.98 \qquad \sigma_T = 0.335 \,. \tag{44}$$

The generated events are split evenly into training and test datasets.

For "Data" we use a more complicated version, to demonstrate the ability of `iHOMER` to fit non-standard functional forms of $f(z|m_T^2)$. We sample the lightcone momentum fraction $z$ from a mixed fragmentation function,

$$f_{\text{data}}(z|m_T^2) \rightarrow f_{a_1,a_2,r,b}(z|m_T^2) = (1-r)f_{a_1,b}(z|m_T^2) + r f_{a_2,b}(z|m_T^2)$$

$$\text{with} \qquad a_1 = 0.3 \qquad a_2 = 0.68 \qquad r = 0.5 \,, \tag{45}$$

keeping $b$ and $\sigma_T$ the same as in Eq.(44). Evaluating each $f(z|m_T^2)$ requires calculating its normalization, which we do numerically. This choice of a mixed fragmentation function has the advantage that it provides a dataset beyond the reach of a parametric fit while still being easy enough to implement via importance sampling while minimizing sample inefficiency. Additionally, we can still compute the exact weights to validate the accuracy and calibration of the learned central values and uncertainties of different weights.

To generate these "Data" events without modifying PYTHIA, we sample $2 \times 10^6$ events from a dataset with the standard Lund string fragmentation function with $a = 0.3$ and weight each event by keeping track of the full history (including both rejected and accepted chains) using only a simple `UserHooks` class [6]. These generation weights $w_{\text{gen}}(H)$ are used to compute all expectation values over functions of $x$, $\mathcal{F}(x)$

$$\left\langle \mathcal{F}(x) \right\rangle_{p_{\text{data}}(x)} \to \left\langle w_{\text{gen}}(H) \mathcal{F}(x) \right\rangle_{p_{\text{gen}}(H)} ,$$

where $H = (S_1^{\text{rej}}, \ldots, S^{\text{acc}})$ and $x = \mathcal{O}(S^{\text{acc}})$. This is needed to compute $\left\langle \log C_\theta(x) \right\rangle_{p_{\text{data}}(x)}$ during the training of the Step 1 classifier via Eq.17 and to compute the binned "Data" distributions in all relevant figures. These events are evenly split into training and test data.

The weighted "Data" events do not alter the `iHOMER` implementation beyond this operational distinction, since $w_\theta(x)$ still approximates $w(x) = \frac{p_{\text{data}}(x)}{p_{\text{ref}}(x)}$. However, weighted events decrease the statistical size of the sample. The ratio between the effective and generated sample size is

$$\frac{\left\langle w_{\text{gen}}(H) \right\rangle^2_{p_{\text{gen}}(H)}}{\left\langle w_{\text{gen}}(H)^2 \right\rangle_{p_{\text{gen}}(H)}} = 0.86 , \tag{46}$$

so we do not observe a significant degradation.

For the choice of observables $x$ we follow Ref. [22] and choose a high-level representation of events consisting of 13 shape variables motivated by the Monash tune [55]:

- event-shape observables $1 - T$, $B_T$, $B_W$, $C$, and $D$. Thrust is defined as [56, 57]

$$T = \frac{\sum_i |\vec{p}_i \cdot \vec{n}_T|}{\sum_i |\vec{p}_i|} , \tag{47}$$

with the axis $\vec{n}_T$ chosen to maximize the above expression. It divides the space into two hemispheres, $S_\pm$, which are then used to compute the jet broadening variables $B_\pm$ [58, 59]

$$B_\pm = \frac{\sum_{i \in S_\pm} |\vec{p}_i \times \vec{n}_T|}{2 \sum_i |\vec{p}_i|} \qquad B_T = B_+ + B_-$$

$$B_W = \max(B_+, B_-) . \tag{48}$$

$C$ and $D$ are related to the eigenvalues of the linearized momentum tensor [60, 61]

$$\Theta^{ij} = \frac{1}{\sum_a |\vec{p}_a|} \sum_a \frac{p_a^i p_a^j}{|\vec{p}_a|}, \qquad i, j = 1, 2, 3 , \tag{49}$$

The three eigenvalues of $\Theta^{ij}$ are denoted $\lambda_{1,2,3}$, and

$$C = 3(\lambda_1 \lambda_2 + \lambda_2 \lambda_3 + \lambda_3 \lambda_1)$$
$$D = 27 \lambda_1 \lambda_2 \lambda_3 . \tag{50}$$

- particle multiplicity $n_f$ and charged particle multiplicity $n_{\text{ch}}$; and
- the first three moments of the $|\ln x_p|$ distribution for all visible particles $x_f$ and for the charged particles $x_{\text{ch}}$, where $x_p = 2|\vec{p}|/\sqrt{s}$ is the momentum fraction of a particle obtained by comparing the momentum of the particle $\vec{p}$ and the center of mass of the collision $\sqrt{s}$. This list is not to be confused with the overall collection of observables $x$. We keep the notation $x_f$ and $x_{\text{ch}}$ to follow standard conventions within the literature.

The full set of observables $x$ is

$$x \equiv \Big\{ T, C, D, B_W, B_T, n_f, n_{\mathrm{ch}},$$
$$\langle |\ln x_f| \rangle, \langle (|\ln x_f| - \langle |\ln x_f| \rangle)^2 \rangle, \langle (|\ln x_f| - \langle |\ln x_f| \rangle)^3 \rangle,$$
$$\langle |\ln x_{\mathrm{ch}}| \rangle, \langle (|\ln x_{\mathrm{ch}}| - \langle |\ln x_{\mathrm{ch}}| \rangle)^2 \rangle, \langle (|\ln x_{\mathrm{ch}}| - \langle |\ln x_{\mathrm{ch}}| \rangle)^3 \rangle \Big\} . \tag{51}$$

We use the first three moments of $|\ln x_p|$ to condense these per-event distributions of particle-level variables into a summary statistic of fixed dimension without needing to resort to arbitrary binning of the distributions.

## 5 Results

Following the motivation and the discussions above, we now assess whether `iHOMER`

1. accurately reweights the fragmentation function and high-level observables from simulation to data without bias; and
2. provides meaningful and well-calibrated uncertainties.

As a closure test, we also demonstrate that the implicit fragmentation functions defined by `HOMER` can be recovered explicitly.

We train `iHOMER` iteratively in the "refinement" style and with uncertainties, as described in Section 3, using the datasets introduced above. The specific training setup and hyperparameters are detailed in App. B. Figure 2 shows the mean sampled AUC score of each Step 1 BNN over each iteration. We show 10 independent `iHOMER` runs, which are essentially indistinguishable because the AUCs are stable to within precise margins; the BNN uncertainties on each point are in the fourth decimal place. Since the AUC closely approaches 0.5, our stopping criterion may not always activate in the same iteration. The histogram overlaid in grey shows the distribution of iterations selected by our statistical AUC criterion, giving an almost even split between iterations 3 and 4. While the results do not depend strongly on the stopping iteration, those runs which stopped in iteration 4 perform marginally better. This is likely due to

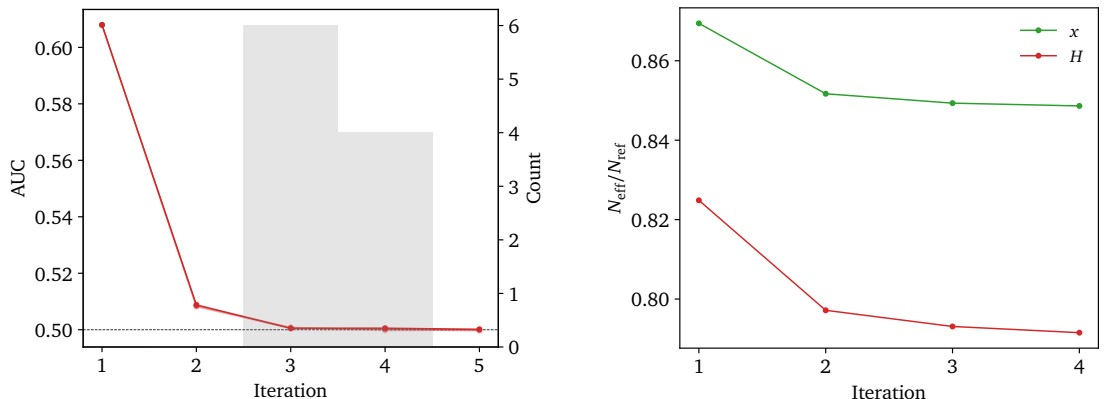

Figure 2: **(Left) Step 1 classifier AUCs over iterations**. For 10 independent runs, the points show the BNN-averaged value and (negligible) associated uncertainty estimated from 10 BNN samples in the test dataset. The gray histogram shows the iterations selected by the stopping criterion. **(Right) Relative effective sample size over iterations**, computed for event- and history-level `iHOMER` weights.

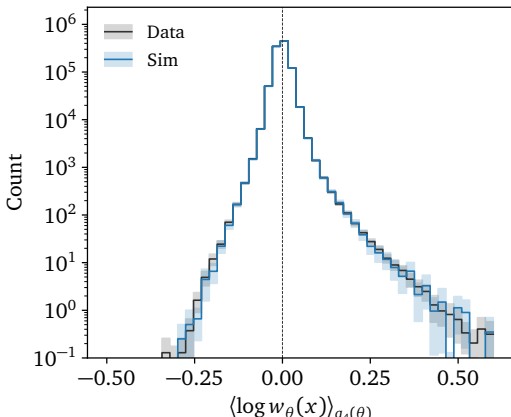

Figure 3: **Step 1 log-weight distributions** for iteration 4. The histogram count and uncertainty comes from computing mean and standard deviation of 10 BNN samples, respectively, allowing count values below 1.

the aforementioned shortcomings of the AUC metric. To be conservative, we randomly select a single run among those that stop in iteration 3 for all the results shown below.

We also keep track of the effective sample size defined in Eq.(41) over iterations for history- and event-level weights in Figure 2. After the first iteration, we observe a clear decrease in the relative effective sample size. Over the next two iterations, however, the relative effective sample size decreases only by 2–3%, revealing that iterating does not dramatically reduce the statistical power of the data.

The classifier weight distribution for the fourth iteration of a single run is shown in Figure 3. These weights are of particular interest because the stopping criterion flags this classifier as random, halting iterations. Visual inspection confirms that the classifier does not detect any local failures that could be missed by the AUC.

## 5.1 Accuracy

To see the impact of the iHOMER iterations we compare the reweighting performance after one (iHOMER-1) and three (iHOMER-3) iterations. The former corresponds to standard HOMER [20], modulo differences in training due to the different Step 2 loss functions. Figure 4 shows the results for select high-level observables, where we include the ratio to the exact reweighting of the reference simulation. This allows easier comparison of different reweightings of the simulation, since statistical fluctuations are shared. "Data" should be distributed around a ratio of 1. The $\chi^2/N_{\mathrm{bins}}$ values against "Data" are shown in the legend. The different distributions shown are the following.

- **Sim**: the reference distributions obtained using the baseline PYTHIA model.
- **Data**: the measured distributions mimicked by synthetic data from the modified mixture model in Eq.(45). One goal of HOMER is to reproduce the "Data" distributions.
- **Exact**: the distributions obtained by reweighting the simulation dataset "Sim" with the exact weights [62]. The "Exact" and "Data" distributions should be identical, up to the increased statistical uncertainties introduced by the reweighting.
- **iHOMER-$n$**: the iHOMER results after the $n$-th iteration. These distributions are obtained by reweighting the simulation dataset "Sim" with the history weights $w_\phi(H)$ defined in Eq.(31).
- **Fit**: To highlight the flexibility of HOMER, which makes no assumptions about the form of the

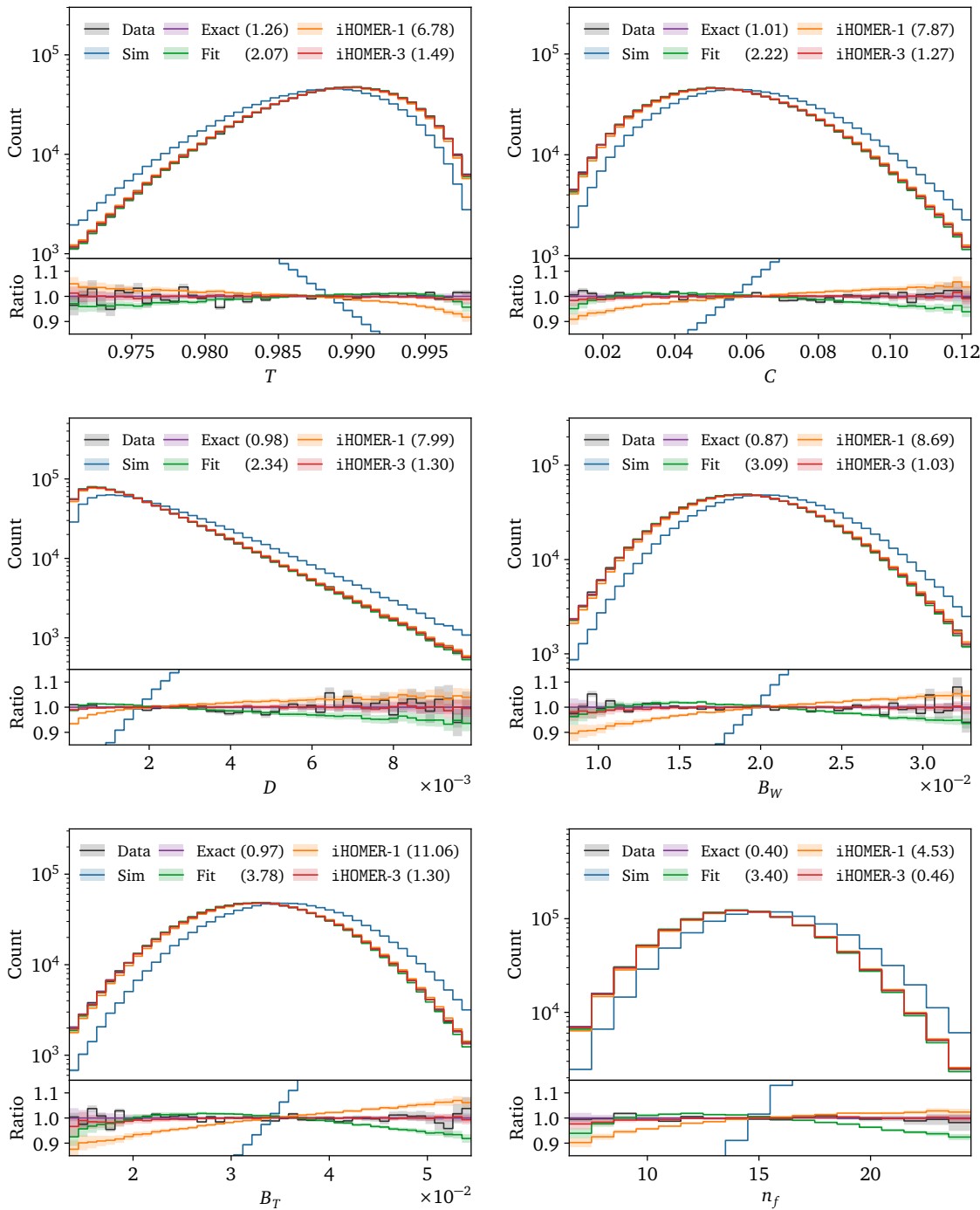

Figure 4: **High-level observable reweighting**, comparing iHOMER with a naive parametric fit. The bottom subpanel shows the ratio to the exact reweighting of the simulation, but the $\chi^2/N_{\text{bins}}$ values indicated in the legend evaluate the agreement with "Data".

fragmentation function, we include a prediction that assumes "Data" is generated using the standard Lund fragmentation function, Eq.(4) rather than the mixed model. This "naive" fit directly uses a set of string breaks for $m_T^2 \in [0.024, 0.029)$ of the "Data" sample. This maximizes the available statistics while fitting a single $f(z|m_T^2)$ without integrating over $m_T$. We perform a binned parametric fit on the $z$ distribution using the iminuit [63] interface to the Minuit minimizer [64] and the parametric form for $f(z|m_T^2 = 0.027)$ with $a$ as a

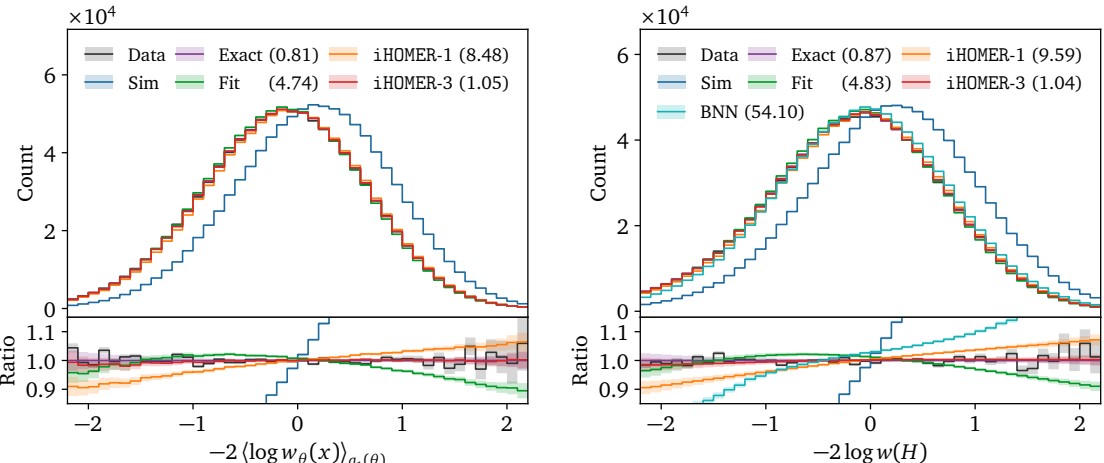

Figure 5: **Optimal observable reweighting**, comparing `iHOMER` with a naive parametric fit at the event (left) and history (right) levels. The subpanel shows the ratio to the exact reweighting of the simulation, while the $\chi^2/N_{\text{bins}}$ values in the legend indicate the agreement with "Data". The optimal event observable is approximated by the mean BNN log-weight from iteration 1.

free parameter. This best-case scenario provides an upper limit on how well a constrained parametric fit can infer the parametric model, since we do not have access to the break-level information in real data. We obtain

$$a_{\text{fit}} = 0.461 \pm 0.006, \tag{52}$$

and use the central value to define break-level weights with Eq.(21) for all $m_T^2$ values. The induced history-level weights provide an additional reweighting of the reference simulation.

In Figure 4 we see that for all observables the `HOMER` prediction is visibly improved over iterations; while `iHOMER-1` deviates from the "Exact" distributions by up to 10%, `iHOMER-3` learns the reweighting at the percent level. Small deviations from closure are only visible in low-statistics regimes, namely in the tails of the event-level distributions. They are typically covered by the uncertainties from `HOMER` and "Data". Since the naive fit assumes the incorrect fragmentation function, we observe almost 10% deviations of "Fit" from "Data", similar to `iHOMER-1`. Although `iHOMER-1` and the parametric fit are clearly biased, these biases are different in nature. `iHOMER-3` matches "Data" better than the parametric fit across all high-level observables.

To compare the power of each set of weights, we examine the optimal observables at the event and history levels, given by $-2\log w(x)$ and $-2\log w(H)$ respectively, in Figure 5. To compute the event-level optimal observable, we use the Step 1 BNN from the first iteration as the estimated event-level likelihood ratio $w(x)$ following Eq.(19), and compare the various reweightings of the "Data" baseline. These optimal observables provide a holistic approach to validate the learned fragmentation function. A low $\chi^2/N_{\text{bins}}$ value when comparing the reweighted simulation to "Data" suggests that the weights capture all relevant information contained across phase space. Additionally, we can use the optimal observables to quantify the information gap between the measured high-level observables and the inaccessible fragmentations. To do so, in the right panel in Figure 5 we show an additional "BNN" distribution. It is obtained by reweighting the simulation dataset "Sim" by the event-level weights as sampled Step 1 BNN from the first iteration, capturing the degree of information encoded in the high-level observables.

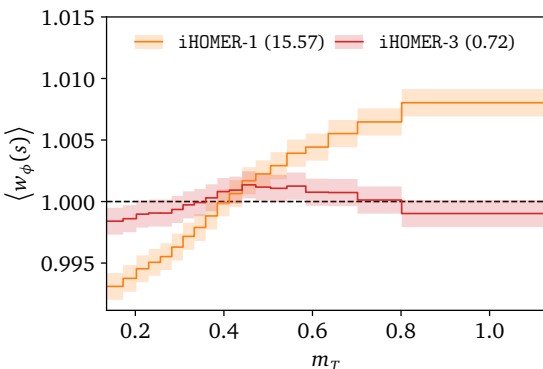

Figure 6: **Conditional normalization** of the fragmentation-level `iHOMER` weights across various $m_T$ bins, comparing the first and final iterations. The $\chi^2/N_{\text{bins}}$ values in parentheses indicate the agreement of the bin values with 1.

For both optimal observables, the `iHOMER-1` and "Fit" weights show a systematic offset with respect to the truth distributions, while `iHOMER-3` reaches almost perfect closure, mimicking the exact weights. Additionally, `iHOMER` generalizes beyond the high-level observables distributions, implied by the "BNN" weights yielding an inferior performance when compared to the "Fit" and `iHOMER`. This highlights the fact that `iHOMER` is actually learning a fragmentation model.

Next, we check whether the learned `iHOMER` weights are normalized to unity in each $m_T$ bin in Figure 6. That is, we compute

$$
\langle w_\phi(s) \rangle_{s \sim p_{\text{ref}}(s), m_T^2 \in [\alpha, \beta]} = \frac{\langle \mathcal{H}(m_T^2 - \alpha)\mathcal{H}(\beta - m_T^2)w_\phi(s) \rangle_{s \sim p_{\text{ref}}(s)}}{\langle w_\phi(s) \rangle_{s \sim p_{\text{ref}}(s)}}, \tag{53}
$$

with $\mathcal{H}(x)$ the Heaviside step function. The closer the learned weights approximate the exact weights, the more accurately they should satisfy the conditional normalization for each $m_T^2$ followed by the exact weights in Eq.(21). Consequently, verifying normalization in this closure test, where the true conditional structure is known, serves as a check of optimality. From Figure 6 we see that the `iHOMER-1` weights do not fully satisfy the conditional normalization, however it is satisfied by the 3rd iteration. Unlike the original `HOMER` implementation in Ref. [22], our method does not require any secondary regularization loss to enforce this structure. Instead, the necessary normalization emerges solely through the optimization of the learned fragmentation function.

Finally, we show the fragmentation function reweighting in Figure 7. In the top row, we show the inclusive distribution of $z$ over all string breaks in the "Data" and "Sim" test datasets, as well as for the various reweightings. Since the break-level weights are defined as ratios of conditional probabilities according to Eq.(13), we need to supplement them with an estimate of $p_{\text{data}}(m_T^2)/p_{\text{ref}}(m_T^2)$ to compute the marginal density

$$
\begin{aligned}
p_{\text{data}}(z) &\equiv \int_{m_{T,\min}^2}^\infty dm_T^2\, f_{\text{data}}(z|m_T^2)p_{\text{data}}(m_T^2) \\
&= \int_{m_{T,\min}^2}^\infty dm_T^2\, w(s)f_{\text{ref}}(z|m_T^2)\frac{p_{\text{data}}(m_T^2)}{p_{\text{ref}}(m_T^2)}p_{\text{ref}}(m_T^2) \\
&= \left\langle w(s)f_{\text{ref}}(z|m_T^2)\frac{p_{\text{data}}(m_T^2)}{p_{\text{ref}}(m_T^2)} \right\rangle_{p_{\text{ref}}(m_T^2)},
\end{aligned} \tag{54}
$$

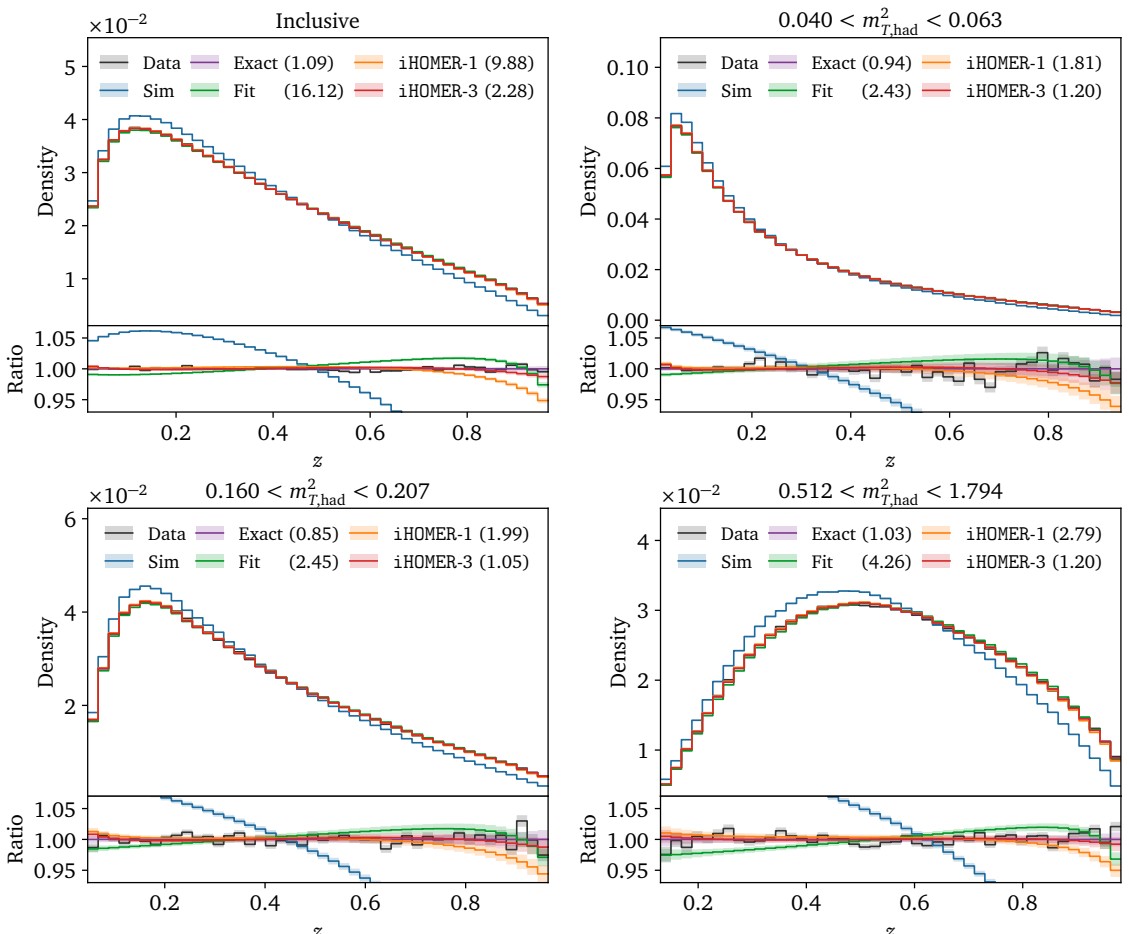

Figure 7: **Fragmentation-level reweighting**, comparing `iHOMER` with a naive parametric fit. The bottom subpanel shows the ratio to the exact reweighting of the simulation, where the $\chi^2/N_{\text{bins}}$ values in parentheses indicated the agreement with "Data". For the inclusive distribution in the top left panel, we supplement the weights with the exact $p_{\text{data}}(m_T^2)/p_{\text{ref}}(m_T^2)$ ratio.

where $m_{T,\text{min}}^2$ is the minimum squared transverse mass. We compute the necessary ratios of probability distributions, $p_{\text{data}}(m_T^2)/p_{\text{ref}}(m_T^2)$, by binning the $m_T^2$ densities of "Data" and "Sim", which is possible in this closure test.

Also in Figure 7 we investigate the behavior of the reweighted fragmentation function in bins of $m_T$ demonstrating that `iHOMER` learns the conditional fragmentation function correctly. The `iHOMER-3` weights reproduce the exact reweighting at the percent level, whereas the fit shows systematic offsets of up to 5%. The only visible deviation between `iHOMER-3` and the exact reweighting is in the tails of the fragmentation function. This shows that `iHOMER` accurately deduces an interpretable reweighting of the fragmentation function, regardless of its analytical form.

## 5.2 Uncertainties

Following the second `iHOMER` improvement, we now evaluate the learned uncertainties. As an initial check, we compare the event-level uncertainty learned in Step 2, $\sigma_\phi(S^{\text{acc}})$, with the spread of the Step 1 log-weights representing the noise on the Step 2 training target, as defined in Eq.(20).

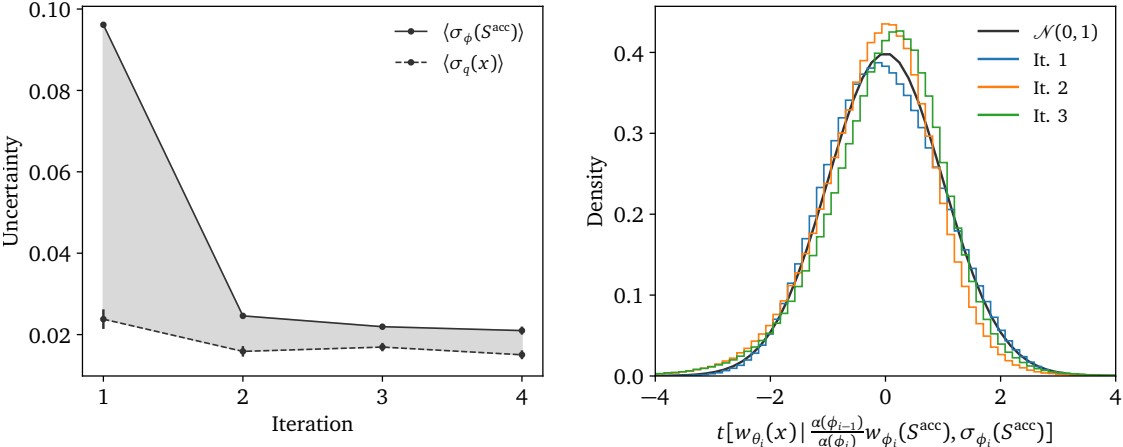

Figure 8: **(Left) Step 1 and Step 2 uncertainties over iterations**. The learned chain-level log-weight uncertainty $\sigma_\phi(S^{\mathrm{acc}})$ is compared to the BNN uncertainty estimated using 10 samples. Each point is the average over the test dataset. The shaded region shows the gap between the two uncertainties, which we broadly refer to as "non-factorization error" in the text. **(Right) Step 2 log-weight calibration** in terms of its pull against the mean classifier log-weight in each iteration, sampling $p_{\mathrm{ref}}^{(i)}(S^{\mathrm{acc}})$ and $q_i(\theta)$.

The left panel in Figure 8 compares iterations 1 to 4. In the first iteration, the learned $\sigma_\phi$ is approximately five times larger than the BNN uncertainty. This suggests that the target noise is not the leading systematic uncertainty in Step 2 during iteration 1. Rather, $\sigma_\phi$ covers the error associated with non-factorization of the classifier $w_\theta(x)$. Over iterations, the learned $\sigma_\phi$ decreases, while the BNN uncertainty is essentially constant after the first iteration. The drop in Step 1 uncertainty from iteration 1 to iteration 2 can be explained by inspecting the AUC evolution in Figure 2. Between the first iterations there is a noticeable drop in AUC, which we expect to be reflected in the statistical uncertainties since the classifier needs to learn a simpler likelihood ratio that can be better constrained with the same, or slightly lower due to $N_{\mathrm{eff}}$, statistics.

The Step 2 uncertainty is always strictly larger than the BNN uncertainty, indicating that Step 2 correctly absorbs the statistical uncertainties captured in Step 1. We attribute the persistent gap to the non-factorization of $w_{\theta_i}(x)$ — since $\alpha(\phi_{i-1})/\alpha(\phi_i) \times w_{\phi_i}(S^{\mathrm{acc}})$ cannot be exactly matched one-to-one with the classifier weight, $\sigma_{\phi_i}(S^{\mathrm{acc}})$ grows to accommodate the error.

To validate this interpretation, we show in the right panel in Figure 8 the calibration of the Step 2 prediction in terms of its pull against the BNN target for each iteration. We construct the pull defined in Eq.(30). For each iteration, the pull follows the unit Gaussian, indicating that the Step 2 uncertainty $\sigma_\phi$ is calibrated. Given the gap between the learned and BNN uncertainties in iteration 1, this correct calibration confirms the presence of a systematic beyond BNN noise, namely the non-factorization error. While other sources of systematic uncertainty can contribute to the Step 2 training, we do not expect them to change in magnitude as much as observed between iterations 1 and 2 in the left panel of Figure 8.

Finally, we compare the network predictions with the exact weights, which represent our primary quantities of interest. For iteration 3, we start with the chain and history results, shown in Figure 9. In both cases, we observe Gaussian but narrow pulls, indicating conservative uncertainties. This is expected if $\sigma_\phi$ contains a large contribution from the non-factorization error; other sources of uncertainty captured by $\sigma_\phi$ should spoil the agreement

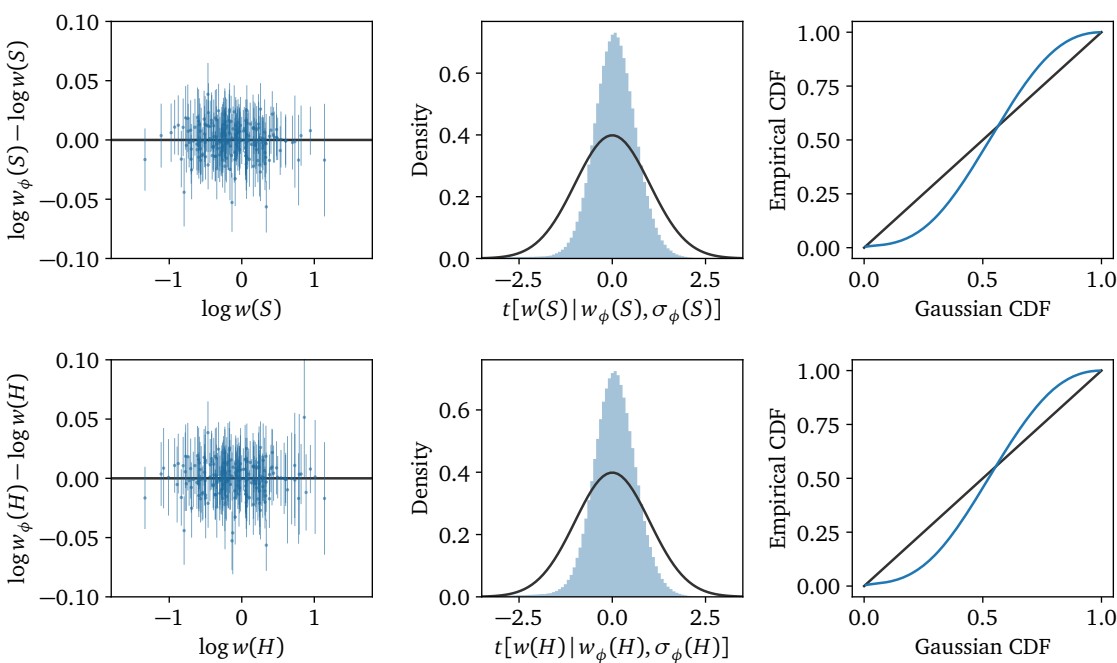

Figure 9: **Chain-level (upper) and history-level (lower) uncertainty calibration**, comparing the predicted log-weight to the exact value. The black curves of the middle and right panels correspond to a unit Gaussian.

with the exact weight. The non-factorization error leads to under-confident predictions because it implies a broad family of fragmentation functions compatible with the observed high-level distributions. We then compare against exact weights that are sampled from a single fixed distribution. In this way, the non-factorization error properly accounts for the degeneracy of the high-level observables.

The calibration of the `iHOMER` uncertainties and its limitations is further illustrated by the calibration of break-level weights shown in Figure 10. We observe how the pulls are overly narrow and also display noticeable non-gaussianity. We further investigate this by examining different $m_T$ quantiles separately (bottom row of Figure 10) and observing how certain subsets of pulls exhibit a more Gaussian behavior than others. The non-Gaussianity can be understood from the fact that the observables lack constraining power on the fragmentation function, and `HOMER` may fail to properly reweight some regions of phase space accordingly. However, the average pull is still null, the associated uncertainty is large, and the resulting event-, chain- and history-level weights are Gaussian, albeit under-confident. This means the `iHOMER` uncertainties properly reflect the lack of knowledge on the fragmentation function due to the high-level observables and the factorization approximation used during training.

To validate the interpretation of the non-factorization error leading to under-confident uncertainties, we can inspect the pull

$$t\left[\frac{\alpha_{\text{ref}}}{\alpha_{\text{data}}}w(S^{\text{acc}})\,\Big|\,\frac{\alpha_{\text{ref}}}{\alpha(\phi)}w_\phi(S^{\text{acc}}),\sigma_q(x)\right]=\frac{\log\frac{\alpha_{\text{ref}}}{\alpha_{\text{data}}}w(S^{\text{acc}})-\log\frac{\alpha_{\text{ref}}}{\alpha(\phi)}w_\phi(S^{\text{acc}})}{\sigma_q(x)}\,.\qquad(55)$$

It tests the coverage of the exact log-weight when assigning the BNN uncertainty to the Step 2 prediction $\log w_\phi(S^{\text{acc}})$. The true value is the event-level weight built using the factorization approximation but considering the exact chain weight $w(S^{\text{acc}})$ and the exact acceptance ratio $\alpha_{\text{ref}}/\alpha_{\text{data}}$. In Figure 11 we show that the BNN component of the uncertainty alone is initially

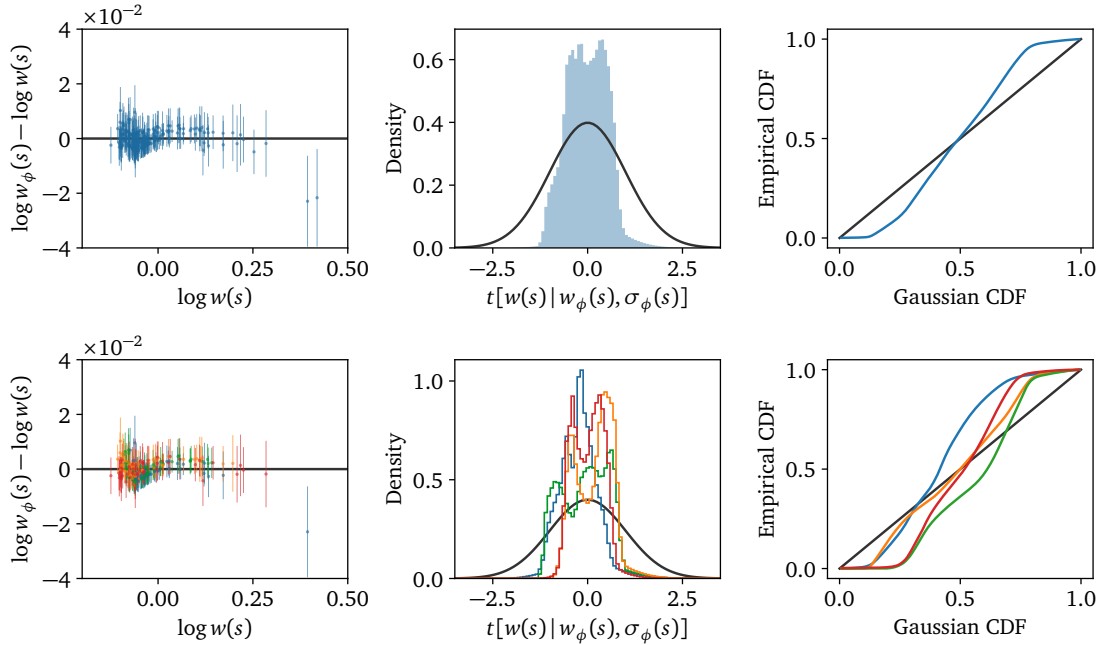

Figure 10: **Fragmentation-level uncertainty calibration**, comparing the predicted log-weight to the exact value. The top row is the inclusive result, while the bottom row is split by quartiles of $m_T$, distinguished by color.

overconfident in iteration 1, but becomes well calibrated by iteration 3. This confirms that if Step 2 could exactly learn the BNN uncertainty, the full calibration shown in Figure 9 would be correct. Fortunately, this effect of non-factorization error only leads to under-confidence, and accounts for the lack of constraining power of the observables on the fragmentation function.

## 5.3 Parameter closure

Although HOMER does not assume any parametric form of the fragmentation function, it can still be related to a physical hadronization model, if needed. As a final check, we perform

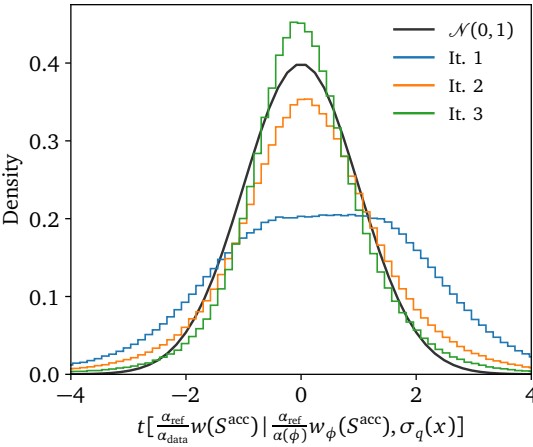

Figure 11: **Calibration of the BNN uncertainty** in terms of the pull between the Step 2 log-weight and the exact value, sampling $p_{\mathrm{ref}}^{(i)}(S^{\mathrm{acc}})$ and $q_i(\theta)$.

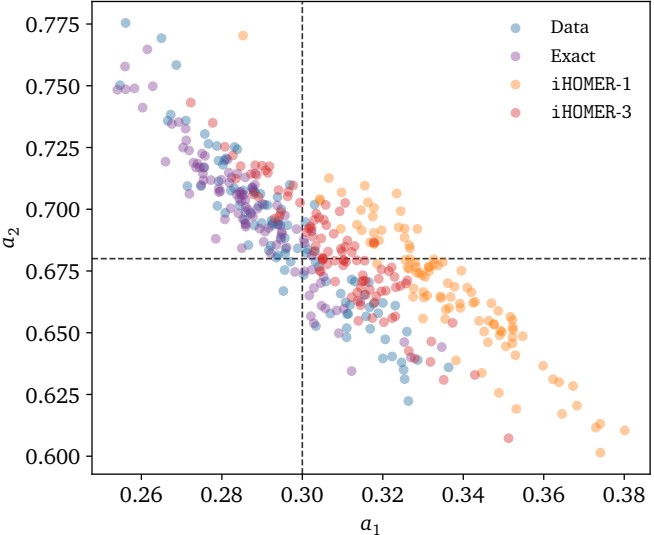

Figure 12: **Ensemble of Maximum Likelihood Estimates of the fragmentation function parameters**, obtained via an unbinned parametric fit on a set of bootstrapped samples. We compare different iterations of `iHOMER`, "Exact" and "Data". True values are shown as dashed lines.

a parametric fit on the learned fragmentation function, using the mixture model defined in Eq.(45) with $a_1$ and $a_2$ as free parameters. To do so, we bootstrap the weighted samples of $z$ for a fixed, narrow $m_T^2 \in [0.024, 0.029)$ bin, again chosen to maximize the statistics while ensuring we only need to compute a single per-class normalization for each set of explored $(a_1, a_2)$ values. We then fit the function

$$f_{a_1,a_2,r=0.5}(z|m_T^2 = 0.027) \tag{56}$$

to the density of $z$ values. For `iHOMER`, we sample weights adding Gaussian noise to $\log w_\phi(s)$ according to the predicted uncertainty $\sigma_\phi(s)$. Unlike the earlier naive fit, this fit is unbinned, to avoid any increased uncertainties due to binning. To accurately find the maximum likelihood estimate (MLE), we implement an expectation maximization algorithm that takes advantage of the mixture model structure by introducing a latent variable representing a class assignment.

In Figure 12, we show an ensemble of 100 MLEs for bootstrapped samples from `iHOMER-1`, `iHOMER-3`, "Exact" and "Data". We use them to obtain a notion of uncertainty on the MLE estimate, although it is not a rigorous analysis. Indeed, `iHOMER-3` improves on `iHOMER-1` and becomes more similar to the "Exact" and "Data", containing the true values in a reasonable region. This means `iHOMER-3` learns something consistent with the true fragmentation function, even given the non-gaussian pulls from Figure 10. It allows us to effectively reweight high-level observables in simulation to data, while maintaining a physical description of hadronization.

# 6  Outlook

We introduced `iHOMER`, an extension of the original `HOMER` method [22,23] that improves its accuracy and precision. Inspired by unfolding techniques [65], the accuracy of `HOMER` is improved through an iterative training which allows us to correct for the invertible observables approximation used during Step 2 training. A data-driven stopping criterion avoids over-iterating

once the reweighting is sufficiently accurate.

The precision is quantified by extending Step 1 and Step 2 to account for different sources of uncertainty. In Step 1, a Bayesian Neural Network quantifies the statistical uncertainties arising from finite data. This uncertainty is propagated to Step 2, where a learnable uncertainty is trained via a heteroscedastic Gaussian loss to capture the dominant systematic uncertainty that prevents exact matching of the Step-1 weights.

To showcase the utility of these two new features, we go beyond fitting `HOMER` to the same Lund fragmentation function with changed parameters and instead consider a more arbitrary fragmentation function to generate the pseudo-data. This highlights how the flexibility of the ML-approach captures non-trivial deviations from the reference function. In this closure test, and using high-level observables to represent the events, we have shown that the iterations increase the performance of `HOMER` and the resulting chain-level and event-level weights are reproduced correctly. We have also inspected the learned fragmentation function and shown via a parametric fit that, although the high-level information cannot perfectly constrain the learned function, it is compatible with the true, known fragmentation function.

Currently, `iHOMER` has been tested on the simplified $q\bar{q}$ scenario. The introduction of gluons presents many challenges, including the variability of the initial state and the decrease in constraining power of the high-level observables. Future work will explore how iterations can improve `HOMER` in those cases, and extend the uncertainty quantification techniques presented here. It should also explore how uncertainty in detector effects and the modeling of perturbative physics (such as the hard processes and the parton shower) should be taken into account, both in terms of accurate modeling and uncertainty quantification.

## Acknowledgments

AO and TP acknowledge support by the Deutsche Forschungsgemeinschaft (DFG, German Research Foundation) under grant 396021762 – TRR 257: *Particle Physics Phenomenology after the Higgs Discovery*, and through Germany's Excellence Strategy EXC 2181/1 – 390900948 (the *Heidelberg STRUCTURES Excellence Cluster*). AB and SPS are supported by the BMBF Junior Group Generative Precision Networks for Particle Physics (DLR 01IS22079). AB acknowledges the continuous support from LPNHE, CNRS/IN2P3, Sorbonne Université and Université de Paris Cité. AY, BA, JZ, MS, and TM acknowledge support in part by the DOE grant DE-SC1019775, and the NSF grants OAC-2103889, OAC-2411215, and OAC-2417682. SM is supported by the Fermi Research Alliance, LLC under Contract No. DE-AC02-07CH11359 with the U.S. Department of Energy, Office of Science, Office of High Energy Physics. CB acknowledges support from the Knut and Alice Wallenberg foundation, contract number 2017.0036. PI and MW are supported by NSF grants OAC-2103889, OAC-2411215, OAC-2417682, and NSF-PHY-2209769. The authors acknowledge support by the state of Baden-Württemberg through bwHPC and the German Research Foundation (DFG) through grant no INST 39/963-1 FUGG (bwForCluster NEMO). This work is supported by the Visiting Scholars Award Program of the Universities Research Association. This work was performed in part at Aspen Center for Physics, which is supported by the NSF grant PHY-2210452.

## Code availability

The code for this paper can be found at https://github.com/ayo-ore/iterative-homer.

# A    Further details on the `HOMER` method

In this appendix we give further details about the `HOMER` method. First, Table 1 contains a translation between the present paper and the notation used in Ref. [23].

Next we give a detailed derivation of Eq.(11), which takes into account the effects of rejected chains through the change of corresponding acceptance rates. In the derivation we need to take into account that each simulated event is not associated with a single chain $S$, but rather to a history $H$ containing a sequence of $N + 1$ chains, $N$ of which were rejected by the `finalTwo` algorithm, and one accepted chain. The probability density for a history for a model $\mathcal{M}$ (in our case, $\mathcal{M} = \{\text{ref}, \text{data}\}$) is the product of accepted and rejected chain densities

$$
p(H|\mathcal{M}) = p(S^{\text{acc}}|\mathcal{M}) \prod_{i=1}^{N} p(S_i^{\text{rej}}|\mathcal{M}) . \tag{57}
$$

By construction, the above probability is normalized over histories of varying lengths

$$
\int dS^{\text{acc}} \sum_{N=0}^{\infty} \int dS_1^{\text{rej}} \cdots dS_N^{\text{rej}} p(H|\mathcal{M}) = 1 . \tag{58}
$$

The corresponding event probability is given by

$$
\begin{aligned}
p(x|\mathcal{M}) &= \int dS^{\text{acc}} \delta(x - \mathcal{O}(S^{\text{acc}})) \sum_{N=0}^{\infty} \int dS_1^{\text{rej}} \cdots dS_N^{\text{rej}} p(H|\mathcal{M}) \\
&= \int dS^{\text{acc}} \delta(x - \mathcal{O}(S^{\text{acc}})) p(S^{\text{acc}}|\mathcal{M}) \sum_{N=0}^{\infty} \prod_{n=1}^{N} \left( \int dS_n^{\text{rej}} p(S_n^{\text{rej}}|\mathcal{M}) \right) \\
&= \int dS^{\text{acc}} \delta(x - \mathcal{O}(S^{\text{acc}})) p(S^{\text{acc}}|\mathcal{M}) \sum_{N=0}^{\infty} (1 - \alpha_{\mathcal{M}})^N \\
&= \int dS^{\text{acc}} \delta(x - \mathcal{O}(S^{\text{acc}})) \frac{p(S^{\text{acc}}|\mathcal{M})}{\alpha_{\mathcal{M}}},
\end{aligned} \tag{59}
$$

where

$$
\alpha_{\mathcal{M}} = \int dS \, \mathcal{A}(S) p(S|\mathcal{M}) = \int dS^{\text{acc}} p(S^{\text{acc}}|\mathcal{M}) = 1 - \int dS^{\text{rej}} p(S^{\text{rej}}|\mathcal{M}), \tag{60}
$$

is the acceptance rate for model $\mathcal{M}$. This proves the relation in Eq.(11) in the main text.

`HOMER` expresses the Lund string fragmentation function implicitly in terms of string-break–level weights $w(s)$

$$
w(s) = \frac{p_{\text{data}}(s)}{p_{\text{ref}}(s)}. \tag{61}
$$

That is, multiplying the reference fragmentation function $f_{\text{ref}}(z|m_T^2)$ by the appropriate weight $w(s)$, where $s$ contains the correct $z$ and $m_T$, gives the reweighted fragmentation function $f_{\text{data}}(z|m_T^2)$.

The chain-level and history-level weights are the products of single emission weights,

$$
w(S) = \prod_{s \in S} \frac{p_{\text{data}}(s)}{p_{\text{ref}}(s)} \quad \text{and} \quad w(H) = \prod_{S \in H} w(S) . \tag{62}
$$

|                       | Ref. [23]                      | This work        |
|-----------------------|--------------------------------|------------------|
| String break          | $\vec{s}_{hcb}$                | $s$              |
| Fragmentation chain   | $\vec{S}_{hc}$                 | $S$              |
| Fragmentation history | $\vec{S}_{h}$                  | $H$              |
| Event                 | $e_h$                          | $x$              |
| Event representation  | $\vec{x}_h$                    | $x$              |
| Acceptance            | $p_i^{\text{acc}}$             | $\alpha_i$       |
| Classifier weight     | $w_{\text{class}}(e_h)$        | $w_\theta(x)$    |
| String break weight   | $w_s^{\text{infer}}(\vec{s}_{hcb}, \theta)$ | $w_\phi(s)$ |
| Chain-level weight    | $w_{\text{infer}}(\vec{S}_{hc}, \theta)$ | $w_\phi(S)$  |
| History-level weight  | $w_{\text{HOMER}}(\vec{S}_{h}, \theta)$ | $w_\phi(H)$   |

Table 1: Translation of the notation used in Ref. [23] (2nd column) to the one used in this work (3rd column).

The event-level weights are obtained by averaging over histories that yield the same event. Written in terms of accepted chains we have,

$$w(x) = \frac{\alpha_{\text{ref}}}{\alpha_{\text{data}}} \langle w(S^{\text{acc}}) \rangle_{p(S^{\text{acc}}|x)}, \tag{63}$$

where the averaged weight for accepted chains given event $x$ is computed as

$$\langle w(S^{\text{acc}}) \rangle_{p(S^{\text{acc}}|x)} = \frac{\int dS^{\text{acc}} \delta(x - \mathcal{O}(S^{\text{acc})}) w(S^{\text{acc}}) p_{\text{ref}}(S^{\text{acc}})}{\int dS^{\text{acc}} p_{\text{ref}}(S^{\text{acc}})} \frac{\int dS^{\text{acc}} p_{\text{ref}}(S^{\text{acc}})}{\int dS^{\text{acc}} \delta(x - \mathcal{O}(S^{\text{acc})}) p_{\text{ref}}(S^{\text{acc}})}$$

$$= \frac{\int dS^{\text{acc}} \delta(x - \mathcal{O}(S^{\text{acc})}) w(S^{\text{acc}}) p_{\text{ref}}(S^{\text{acc}})}{\int dS^{\text{acc}} \delta(x - \mathcal{O}(S^{\text{acc})}) p_{\text{ref}}(S^{\text{acc}})}. \tag{64}$$

In practice, we do not perform the above-defined averaging, but rather work in the approximation where $\langle w(S^{\text{acc}}) \rangle_{p(S^{\text{acc}}|x)}$ is replaced by $w(S^{\text{acc}})$ for a single chain $S^{\text{acc}}$ that happened to be produced in the simulation (and is giving event $x$). Effectively, the averaging occurs later in the analysis chain, when one bins the distributions of observables, thus pooling many events $x$. Alternatively, one could perform explicit averaging over neighboring events, as was done in [23].

The acceptance rate $\alpha_{\text{data}}$ in Eq.(63), describing acceptance of chains in simulation of hadronization using the true fragmentation function $f_{\text{data}}(z|m_T^2)$, is given by

$$\alpha_{\text{data}} = \int dS^{\text{acc}} p_{\text{data}}(S^{\text{acc}}) = \int dS^{\text{acc}} w(S^{\text{acc}}) p_{\text{ref}}(S^{\text{acc}}), \tag{65}$$

and estimated in a similar fashion to Eq.(12),

$$\alpha_{\text{data}} = \frac{\sum_{m=1}^{M} w(S^{\text{acc}})}{\sum_{m=1}^{M} \sum_{S \in H} w(S)}. \tag{66}$$

In `HOMER`, the string break-level weights $w(s)$ in Eq.(61) are parameterized by a neural network with parameters $\phi$. This then enters in the r.h.s. of Eq.(63) via $w(S^{\text{acc}})$ and the acceptance ratio $\alpha_{\text{data}}$. The values of NN parameters $\phi$ are then optimized to match the event weight $w(x)$ obtained in Step 1 of `HOMER` (see Section 3).

|  | Step 1 | Step 2 |
|---|---|---|
| MLP hidden channels | 2 | 3 |
| MLP hidden layers | 32 | 64 |
| MLP activation | SiLU | SiLU |
| $N_{\text{data}}$ (training/validation) | 900k/100k | - |
| $N_{\text{ref}}$ (training/validation) | 900k/100k | 900k/100k |
| Loss | ScheduleFree AdamW | ScheduleFree AdamW [66] |
| Learning rate | $10^{-4}$ | $10^{-3}$ |
| Weight decay | None | None |
| Batch size | 8192 | 8192 |
| Epochs | 500 | 2500 |
| Patience | None | 250 |

Table 2: Network and optimization hyperparameters for training networks in `iHOMER`. Both MLPs in Step 2 share the same architecture.

## B  Training details and hyperparameters

In this appendix we give further details about the BNNs used and the training. The complete list of network and optimization hyperparameters are given in Table 2.

Before training, we normalize each high-level observable and string-break vector by subtracting the mean and dividing by the standard deviation of the full training dataset. During Step 2, we find it beneficial to explicitly normalize the target weights using

$$w_{\theta_i}(x) \rightarrow w_{\theta_i}(x)/\left\langle w_{\theta_i}(x)\right\rangle_{p_{\text{ref}}^{(i)}(S^{\text{acc}})}, \tag{67}$$

which uses paired events $x = \mathcal{O}(S^{\text{acc}})$. Eq.(67) applies to both iteration styles, so long as the correct $p_{\text{ref}}^{(i)}$ is used. This helps to prevent reinforcement of mis-normalized weights over iterations.

Since the Bayesian loss in Eq.(17) consists of two terms, the likelihood loss and the KL-regularization, it is important to track them separately. Here, a relevant hyperparameter is the width of the Gaussian prior $p(\theta)$. We confirmed that all presented results are consistent under a wide range of different priors, while for the numerical results shown in this paper we chose a prior width of 1. If the prior is chosen too narrow, the BNN posterior will not sufficiently cover $\theta$ space.

## C  BNN results

In this appendix we show the reweighting results of the Step 1 BNN classifier itself, which is trained as specified in App. B.

Figure 13 shows the distributions of high-level observables for "Data" and "Sim", as well as for the simulation reweighted either by the BNN, or by the exact weights (denote as "Exact") in the figure. The uncertainty bands for "Data", "Sim" and "Exact" are standard (weighted) Poisson statistics per bin $b$

$$\sigma_b^2 = N_{\text{ref},b}\left\langle w^2(x_i)\right\rangle_{i\in b} = \sum_{i\in b} w^2(x_i)\,, \tag{68}$$

with unit weights for "Data" and "Sim". The "BNN" uncertainty includes the variance from

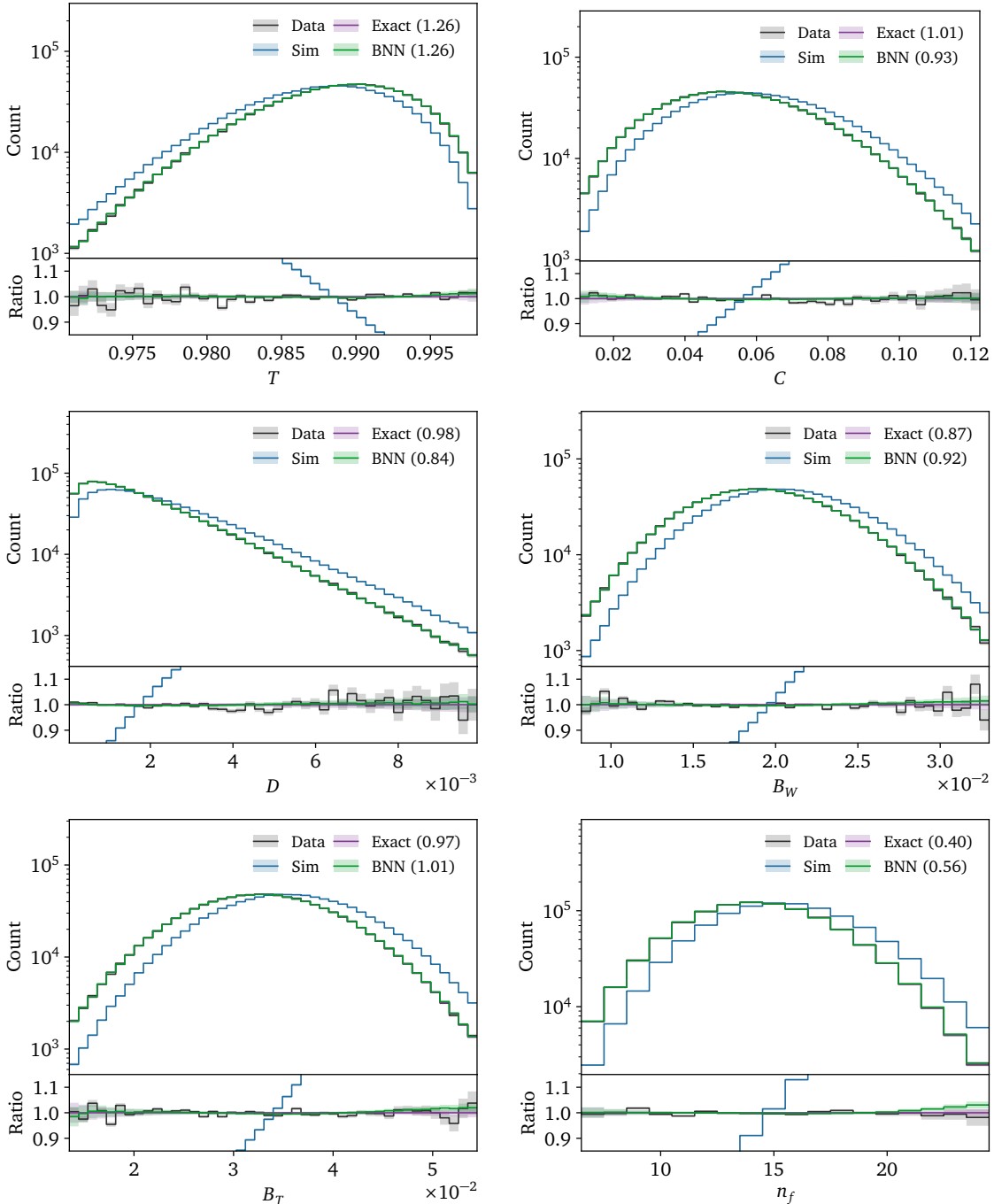

Figure 13: **High-level reweighting with Step 1 classifier**: Distributions of select high-level observables for data and simulation, as well as and simulation reweighted by either classifier weights or exact weights. The classifier error band includes sampling over the BNN parameter posterior.

sampling $\theta \sim q(\theta)$,

$$\sigma_{q,b}^2 = N_{\mathrm{ref},b} \left\langle w_\theta^2(x_i) \right\rangle_{i \in b, \theta \sim q(\theta)} = \frac{1}{M} \sum_{j=1}^{M} \sum_{i \in b} w_{\theta_j}^2(x_i) \,. \tag{69}$$

The BNN matches the "Data" distribution precisely throughout all of the observables, with small deviations present only in the low-statistics tails.

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
