# Peer review of "Iterative HOMER with uncertainties"

_SciPost Physics_

## Round 1 · Referee Report · Anonymous (Referee 1) · 2025-10-14

Strengths

1 - Improves both accuracy and precision of extracted fragmentation functions compared with previous versions of the approach.
2 - Contains all necessary details and a detailed demonstration of the performance.
3 - Has the potential for other applications in the field.

Weaknesses

None

Report

Report about arXiv:2509.03592v1

Iterative HOMER with uncertainties

by Anja Butter et al.

A majority of experimental analyses of collider data are currently
based on a comparison with Monte Carlo simulations for which
the LUND hadronization model are used. This requires fragmentation
functions as input which have to be determined from data. The
authors present a method to extract fragmentation functions using
a combination of Bayesian neural networks and regression. The
submitted paper is the third in a series and presents two important
improvements over the previous versions: an iterative procedure
and modifications in the architectural choices lead to a better
assessment of both accuracy and precision for the extracted
fragmentation function.

The paper is well written and contains all necessary details, as well
as a detailed demonstration of the performance achieved. I expect
that the information given in the paper together with the code,
which is made available on github.com, will allow other workers
in the field to benefit also for other applications.

I support publication of the submitted article without modifications.

Requested changes

None

Recommendation

Publish (surpasses expectations and criteria for this Journal; among top 10%)

---

## Round 1 · Referee Report · Anonymous (Referee 2) · 2025-10-17

Strengths

  1. Clear description of the new method.
  2. Proper treatment of uncertainties.
  3. The method is physically interpretable.
  4. Rigorous validation through closure test example.
  5. Publicly accessible code for reproducibility.

Weaknesses

  1. Dependence on a specific hadronization model (string model).
  2. Lack of application to real data (or mismodeled data).
  3. No benchmark on computing time shown.

Report

The paper introduces iHOMER, which is an iterative extension of the HOMER method for extracting the Lund fragmentation function from data. It systematically addresses the information gap between latent and observable phase spaces via iteration. In addition, it incorporates uncertainty quantification using Bayesian Neural Networks and heteroscedastic regression. The method is clearly described in the paper with a concrete example for validation of the method, which shows improved accuracy and well calibrated uncertainties compared with the original HOMER method.

This is a solid and technically sound piece of work. I recommend publication, provided that the authors address a few minor comments.

Requested changes

  1. Do I understand correctly that the method is specific to the string model? If so, could you comment on the limitation on the flexibility coming from the string model? And would it be possible to adapt the method to other hadronization model, e.g. cluster?

  2. The method assumes negligible systematic uncertainty in the reweighing step 1 and negligible statistical uncertainty in the factorization step 2. Is this always guaranteed? What would one need to do if otherwise?

  3. Could you explain further how using the Machine Learning based hadronization models could improve MCEG accuracy?

  4. Could you provide any benchmarks on the training / computing time with the proposed method?

Recommendation

Ask for minor revision

---

## Editorial Decision

awaiting_resubmission